# Min waves without MinC can pattern FtsA-anchored FtsZ filaments on model membranes

Elisa Godino[1], Anne Doerr[1] & Christophe Danelon [1✉]

Although the essential proteins that drive bacterial cytokinesis have been identified, the precise mechanisms by which they dynamically interact to enable symmetrical division are largely unknown. In *Escherichia coli*, cell division begins with the formation of a proto-ring composed of FtsZ and its membrane-tethering proteins FtsA and ZipA. In the broadly proposed molecular scenario for ring positioning, Min waves composed of MinD and MinE distribute the FtsZ-polymerization inhibitor MinC away from mid-cell, where the Z-ring can form. Therefore, MinC is believed to be an essential element connecting the Min and FtsZ subsystems. Here, by combining cell-free protein synthesis with planar lipid membranes and microdroplets, we demonstrate that MinDE drive the formation of dynamic, antiphase patterns of FtsA-anchored FtsZ filaments even in the absence of MinC. These results suggest that Z-ring positioning may be achieved with a more minimal set of proteins than previously envisaged, providing a fresh perspective about synthetic cell division.

[1] Department of Bionanoscience, Kavli Institute of Nanoscience, Delft University of Technology, Delft 2629HZ, The Netherlands. ✉email: c.j.a.danelon@tudelft.nl

n *Escherichia coli* bacteria, a ring-forming multiprotein complex drives membrane constriction at the future division site[1]. The proto-ring is composed of the three proteins FtsZ, FtsA and ZipA[2,3]. FtsZ is a tubulin homolog with GTPase activity that can polymerize into protofilaments[4,5]. FtsZ has no affinity for lipids and is anchored to the membrane by ZipA and the actin homolog FtsA[6–8]. This set of proteins acts as a scaffold, recruiting other factors to form a mature divisome. The Min system in *E. coli* is composed of the three proteins MinC, MinD, and MinE. Together, they provide the localization cues that restrict the assembly of the FtsZ ring to the middle of the cell, following a commonly accepted sequence of molecular events[9,10]. MinD is an ATPase that dimerizes and is recruited on the membrane when bound to ATP. Once at the membrane, MinD interacts with MinE, which stimulates the ATPase activity of MinD leading to the release of both proteins from the membrane. This dynamic interplay between MinD and MinE, together with the geometrical constraints of the cell, cause oscillating gradients of the two proteins at the cell poles[11–14]. MinC travels with the MinDE waves, where it inhibits the assembly of FtsZ protofilaments. As a result of the Min oscillations, the time-averaged concentration of the MinD-MinC complex is higher at the poles, restricting the formation of the Z-ring to mid-cell[15–17].

Despite numerous in vitro studies about membrane-anchored FtsZ[18–21] and Min dynamics[17,22–24], data about the spatio-temporal regulation of FtsA-FtsZ cytoskeletal structures by MinDEC are scarce[25]. It is known that MinC is required for correct placement of the division site in vivo[11,14]. On the other hand, several reports suggest that MinDE oscillations may act as a spatial regulator of membrane proteins[26–28]. The abundances of the membrane proteome in wild-type and Δ*min E. coli* cells were compared, revealing that Min oscillating gradients modulate protein association with the inner membrane[26]. Moreover, MinDE dynamic patterns spatiotemporally regulate and transport peripheral membrane proteins on supported lipid bilayers (SLBs) and in microcompartments[27,28]. These recent findings raise the

question of whether the oscillating MinDE gradients could influence membrane localization of FtsA-anchored FtsZ filaments in the absence of MinC. Interestingly, when FtsZ-YFP-MTS (a chimera of truncated FtsZ, YFP, and the membrane-targeting sequence of MinD) was combined with MinDE proteins on an SLB, static FtsZ-YFP-MTS networks were not affected by the Min surface waves[27]. The use of a chimeric FtsZ in these in vitro assays leaves open questions, since the ability of MinDE oscillating gradients to influence the lateral distribution of FtsZ might be different with the native membrane anchor FtsA.

Herein, we set out to elucidate this question by combining MinDE(C), where '(C)' indicates the presence or absence of MinC, and FtsA-FtsZ in open and closed model membrane systems. As a medium for our cell-free assays, we use the PURE system, a minimal gene expression system reconstituted primarily from *E. coli* constituents[29,30]. The activity of PURE-expressed FtsA and MinDEC from single-gene constructs has already been validated[25,31]. Here, we design a DNA template containing the *E. coli* genes *ftsA*, *minD* and *minE* concatenated in the form of three transcriptional units (tri-cistron). All proteins are expressed at physiologically relevant levels and in active states, allowing us to examine the interplay between the spatial organization of FtsA-FtsZ cytoskeletal structures and MinDE(C) membrane patterns on SLBs and within microdroplets. We find that MinDE surface waves act as a spatial regulator of FtsA-anchored FtsZ filaments in a MinC-independent manner. The possible implications regarding bacterial cell division and implementation of a minimal FtsZ ring in synthetic cells are discussed.

## Results

**Cell-free co-expression of FtsA, MinD and MinE at physiological levels.** As an experimental design strategy to co-reconstitute the Min and FtsZ subsystems, we created a multicistronic DNA template containing the genes of the *E. coli* proteins FtsA, MinD, and MinE (Fig. 1a). The DNA template (5 nM) was constitutively

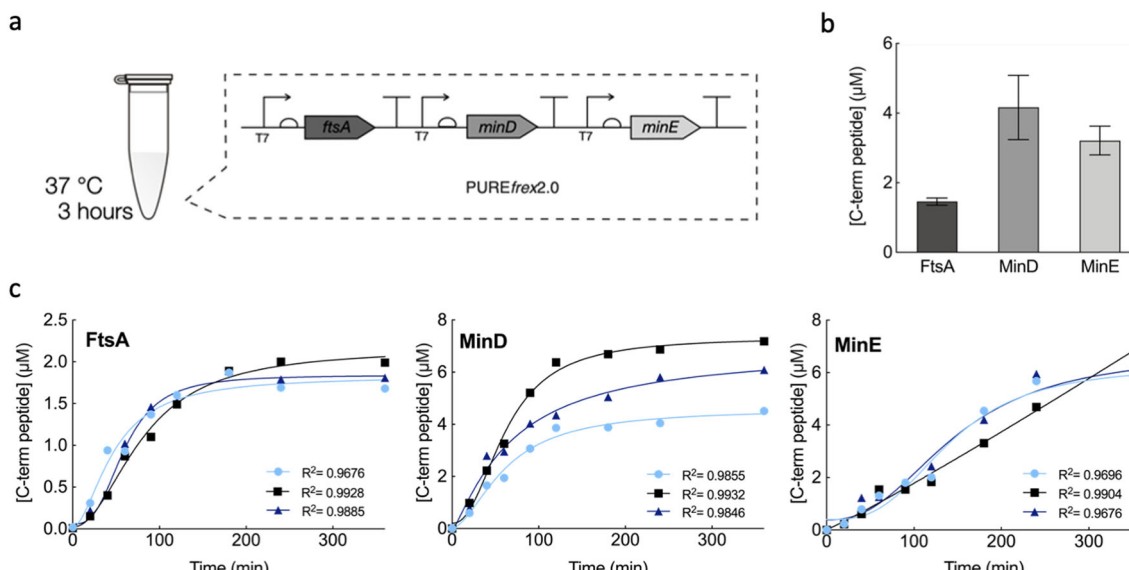

**Fig. 1 Quantification of cell-free expressed FtsA, MinD and MinE proteins from a single DNA construct. a** Each gene, *ftsA*, *minD* and *minE*, was placed under its own T7 promoter, ribosome binding site and T7 transcription terminator and was constitutively expressed with PURE*frex*2.0 in test-tube reactions. **b** Quantitative LC-MS analysis of FtsA, MinD and MinE protein production after 3 h of expression at 37 °C. Abundance of the most C-terminal proteolytic peptide was quantified for each protein using an internal standard (QconCAT). Data represent mean values over three biological repeats, each analyzed in three technical replicates. **c** Concentration of FtsA, MinD and MinE (most C-terminal peptide) as a function of the expression time course (from 0 to 6 h). Symbols represent data from three independent experiments. The solid lines are fits to a mathematical model for gene expression kinetics (Methods section).

expressed in PUREfrex2.0 and the abundance of the three synthesized proteins was quantified by targeted liquid chromatography-coupled mass spectrometry (LC-MS) (Fig. 1a and Fig. S1). After 3 h of expression, the obtained concentrations were 1.5 ± 0.1 μM for FtsA, 4.2 ± 0.9 μM for MinD and 3.2 ± 0.4 μM for MinE (mean ± SD, three biological replicates) (Fig. 1b). These values give a good approximation of the total amounts of full-length proteins but may overestimate the concentrations of active proteins. Similar concentration values were reported in vivo: ~0.5 μM for FtsA[32], and an equimolar amount of MinD and MinE ranging between 1 and 3 μM[33,34]. Kinetics of FtsA, MinD and MinE production revealed that most of FtsA and MinD proteins were synthesized within the first 3 h of co-expression (Fig. 1c). The apparent translation rate was calculated by fitting a phenomenological model to the experimental data yielding values of 0.018 ± 0.002 μM min$^{-1}$ for FtsA, 0.043 ± 0.016 μM min$^{-1}$ for MinD and 0.027 ± 0.002 μM min$^{-1}$ for MinE (mean ± SD of fitted parameter values, three biological replicates). The expression lifespan, defined as the time point at which protein production stops, was determined for each protein: 119 ± 28 min (FtsA), 144 ± 28 min (MinD) and 250 ± 16 min (MinE). These values are consistent with those obtained from kinetic profiles of other PURE-expressed proteins[35,36].

**Integration of MinDEC dynamic patterns and FtsA-FtsZ cytoskeletal structures on SLBs.** We next investigated how the MinDEC and FtsA-FtsZ networks mutually interact on SLBs. The tri-cistron DNA was expressed in test-tube reactions, in the presence of DnaK mix, to produce FtsA and MinDE. Purified FtsZ-A647 (FtsZ conjugated to the AlexaFluor647 fluorophore, 3 μM) and eGFP-MinC (fusion between the enhanced green fluorescent protein and MinC, 0.5 μM) were subsequently added, along with adenosine triphosphate (ATP, extra 2.5 mM) and guanosine triphosphate (GTP, extra 2 mM) (Fig. 2a). Sample was transferred onto an SLB and the membrane organization of the FtsZ- and Min-subsystems was imaged by time-lapse fluorescence microscopy. We found that the membrane area close to the center of the chamber exhibited curved filaments and ring-like structures of FtsA-anchored FtsZ (Fig. 2b), with cytoskeletal properties similar to that observed in the absence of Min proteins[21,31]. In contrast, the edge of the chamber was occupied by MinDEC dynamic patterns only. In between these two regions, mixed patterns involving the two subsystems were observed (Fig. 2b). Spatial discrimination in distinct domains occurred immediately after supplying the PURE sample. We assume that molecular diffusion at the membrane is constrained by the edge of the chamber, resulting in a higher concentration of membrane-bound MinD and MinE[37,38] that outcompete FtsA for membrane coverage.

In the areas where FtsA-FtsZ and Min proteins coexisted, FtsZ could undergo different types of dynamic behaviors shaped by Min oscillatory gradients, including planar waves, rotating spirals, and standing waves (Fig. 2b, c and Fig. S3). Fluorescence intensity profiles showed that membrane localization of MinC and FtsZ anti-correlated (Fig. 2d, Supplementary Movie 1). The intensity distribution found for the traveling wave was broader for FtsZ than MinC. Calculated wavelength of MinC planar waves in the region where the two subsystems coexisted was 113 ± 30 μm and the wave velocity was 0.71 ± 0.30 μm s$^{-1}$ (mean ± SD, from three biological replicates). Standing waves showed a characteristic oscillation time of 105 ± 24 s. In the area where FtsZ signal was not detectable, the Min wavelength was 57 ± 13 μm, i.e. about half the value measured in the intermediate region. This result suggests that membrane-bound FtsA-FtsZ cytoskeletal structures influence the Min wave properties by increasing the

wavelength (Fig. S3b). When omitting FtsA through expression of a bi-cistronic DNA template containing only the minD and minE genes, FtsZ ring-like structures were no longer observed on the bilayer (Fig. 2e, Supplementary Movie 2). Instead, FtsZ traveled in phase with MinC on the MinDE waves (Fig. 2e, f, Supplementary Movie 2). Colocalization of the MinC and FtsZ patterns is consistent with previous results that indicated weak transient interaction between MinC and FtsZ[39–41]. The observation that FtsZ travels with MinC in the absence of FtsA, while being anti-correlated with MinC when FtsA is present, is in agreement with a competition mechanism between MinD-bound MinC and the membrane anchoring protein for the binding to the C-terminal peptide region of FtsZ[42]. Calculated wavelength was 59 ± 14 μm, similar to the value measured for MinDEC patterns in the area where FtsA-FtsZ were excluded. This result supports the hypothesis that the presence of FtsA-FtsZ on the membrane increases the wavelength of propagating Min waves.

Following a traditional view of the interplay between the two subsystems, the dynamic behavior of FtsZ reported in Fig. 2b, c is expected to be the result of the inhibitory interaction of traveling MinC on FtsA-bound FtsZ. However, a different interpretation would be that FtsA-FtsZ oscillatory gradients originated from the direct action of the MinDE proteins, a possible scenario that we investigated in the next section.

**MinDE oscillatory gradients, without MinC, drive the formation of FtsA-FtsZ cytoskeletal patterns on SLBs.** To explore the hypothesis that MinC has a dispensable function in regulating FtsA-FtsZ membrane dynamics, we repeated the experiments described above, this time replacing purified eGFP-MinC by a trace amount of purified eGFP-MinD (100 nM) as a reporter of the Min oscillations (Fig. 3a). On a large scale, FtsA-FtsZ structures were more prominent in the center of the chamber, while Min waves preferentially populate the outer area of the chamber. Coexistence of both MinDE and FtsA-FtsZ dynamic patterns occurred in between these two regions (Fig. 3b), similarly as observed in the presence of MinC (Fig. 2b).

Several dynamic behaviors were found in the extended area in which FtsA-anchored FtsZ ring-like structures and MinDE coexisted. In most fields of view MinDE patterns could effectively rearrange FtsA-FtsZ filaments (Fig. 3c, d, Fig. S4a, Supplementary Movie 3 and 5). In a few cases, anticorrelated propagating waves of FtsZ and MinDE with a low amplitude were also observed (Fig. 3e, f, Supplementary Movie 4). These two types of rearrangements might be the manifestation of different local concentrations of MinDE or FtsA-FtsZ on the bilayer. Also, areas with a higher concentration of FtsA and FtsZ may support formation of more stable cytoskeletal structures (longer residence time on the membrane) that are less susceptible to be redistributed by MinDE. The fact that we did not observe such dim propagating waves of FtsZ in the presence of MinC (Fig. 2b, c) can be explained by MinC's depolymerizing effect that prevents stable FtsZ bundles from forming, reduces the local concentration of FtsA-FtsZ and facilitates propagation of the MinDE diffusion barrier.

Quantifying the Min wave properties in the areas where FtsA-FtsZ cytoskeletal structures were effectively rearranged, we calculated a wavelength for the MinD signal of 76 ± 30 μm and wave velocity of 0.5 ± 0.2 μm s$^{-1}$ (mean ± SD, from three biological replicates), while standing waves had a characteristic oscillation time of 109 ± 36 s. In regions with no visible FtsZ signal, a Min wavelength of 71 ± 22 μm was measured, with no differences between the center and the edge of the chamber (Fig. S4b). In our previous work[25], MinDE surface waves in PURE system exhibited a wavelength of 43 ± 7 μm and a velocity

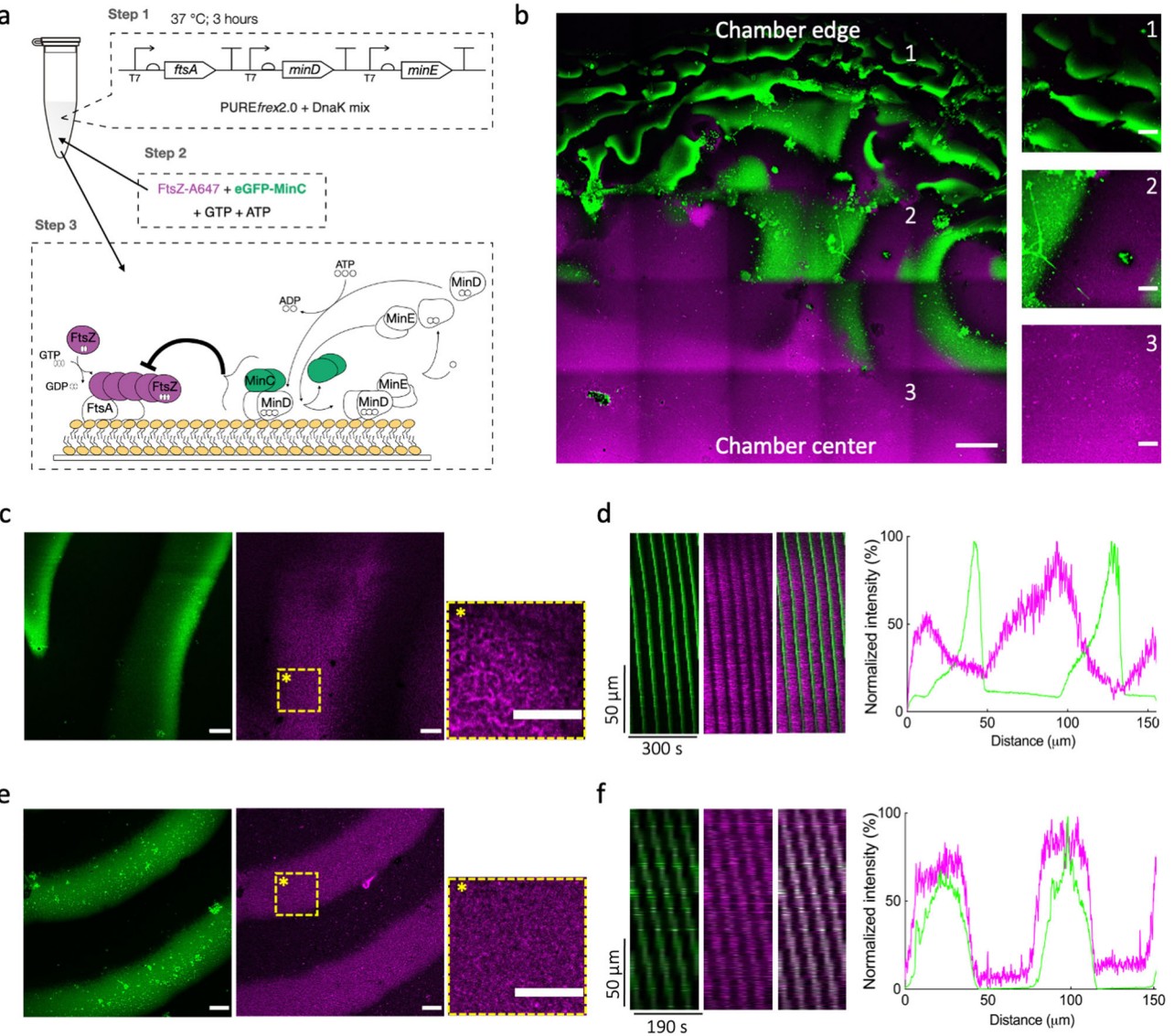

**Fig. 2 Integration of MinDEC surface waves with FtsA-anchored FtsZ filaments on supported membranes. a** Schematic of the experimental approach. Genes *ftsA, minD* and *minE* were co-expressed from the three-cistron DNA template (5 nM) in PURE system with a DnaK chaperone mix for 3 h at 37 °C. The solution was then supplemented with 2.5 mM ATP, 2.0 mM GTP, 0.5 μM eGFP-MinC and 3 μM FtsZ-A647 before transfer on top of an SLB. The cartoon depicts the main biochemical steps underlying FtsZ polymerization, its recruitment to the membrane by FtsA, and the influence of the reconstituted Min network. A schematic of the chamber used in the SLB assays is depicted in (Fig. S2). **b** Mosaic of 5 × 5 tile scan microscope images showing large-scale organization of FtsA-FtsZ and MinDEC dynamic patterns. FtsA-anchored FtsZ filaments mostly populate the center of the chamber (3), while MinDEC oscillating gradients dominate at the edges of the chamber (1). In between these two areas (2), correlated patterns of the two subsystems can be seen. Signals from eGFP-MinC and FtsZ-A647 are in green and magenta, respectively. Composite images of overlaid channels are shown. Scale bar is 50 μm in the mosaic image and 10 μm in the three zoomed-in images. **c** Fluorescence microscopy images of FtsA-FtsZ and MinDEC dynamic patterns taken from the intermediate SLB area (2). A zoom-in image of the framed area in the FtsZ channel is also displayed, showing the organization of FtsA-anchored FtsZ into filaments. Scale bars are 10 μm. **d** The time evolution of planar waves was analyzed and kymographs were constructed. On the right, representative intensity profiles of MinC and FtsZ are shown. Color coding is the same as for the microscopy images. **e** Fluorescence microscopy images of FtsZ and MinDEC dynamic patterns when omitting FtsA. A zoom-in image of the framed area in the FtsZ channel is also displayed, showing that in the absence of FtsA, FtsZ fails to organize into cytoskeletal structures. **f** Kymographs constructed from the time series images shown in (**e**). On the right, intensity profiles of MinC and FtsZ signals showing that FtsZ travels with the MinDEC waves in the absence of a membrane anchor.

of 0.5 ± 0.1 μm s⁻¹. The data reported here show an increased wavelength of the MinDE oscillations, suggesting that FtsA-FtsZ cytoskeletal structures influence the wave properties (Fig. S4b). Another possibility could be the difference in MinDE concentrations in the two studies (4.2 ± 0.9 μM and 3.2 ± 0.4 μM in the present study vs 19 ± 7 μM and 5 ± 4 μM in the previous study, for MinD and MinE respectively). Moreover, we found that the

wavelength increased in the presence of MinC, while the velocity remained mostly unaffected (Fig. 3g, h). Varying the MinE-to-MinD ratio did not to change the Min wave properties nor the re-localization of FtsA-bound FtsZ filaments (Fig. S4d).

We then asked whether FtsA alone, i.e. not engaged in FtsA-FtsZ cytoskeletal structures, was also subjected to spatial reorganization by the MinDE propagating diffusion barrier.

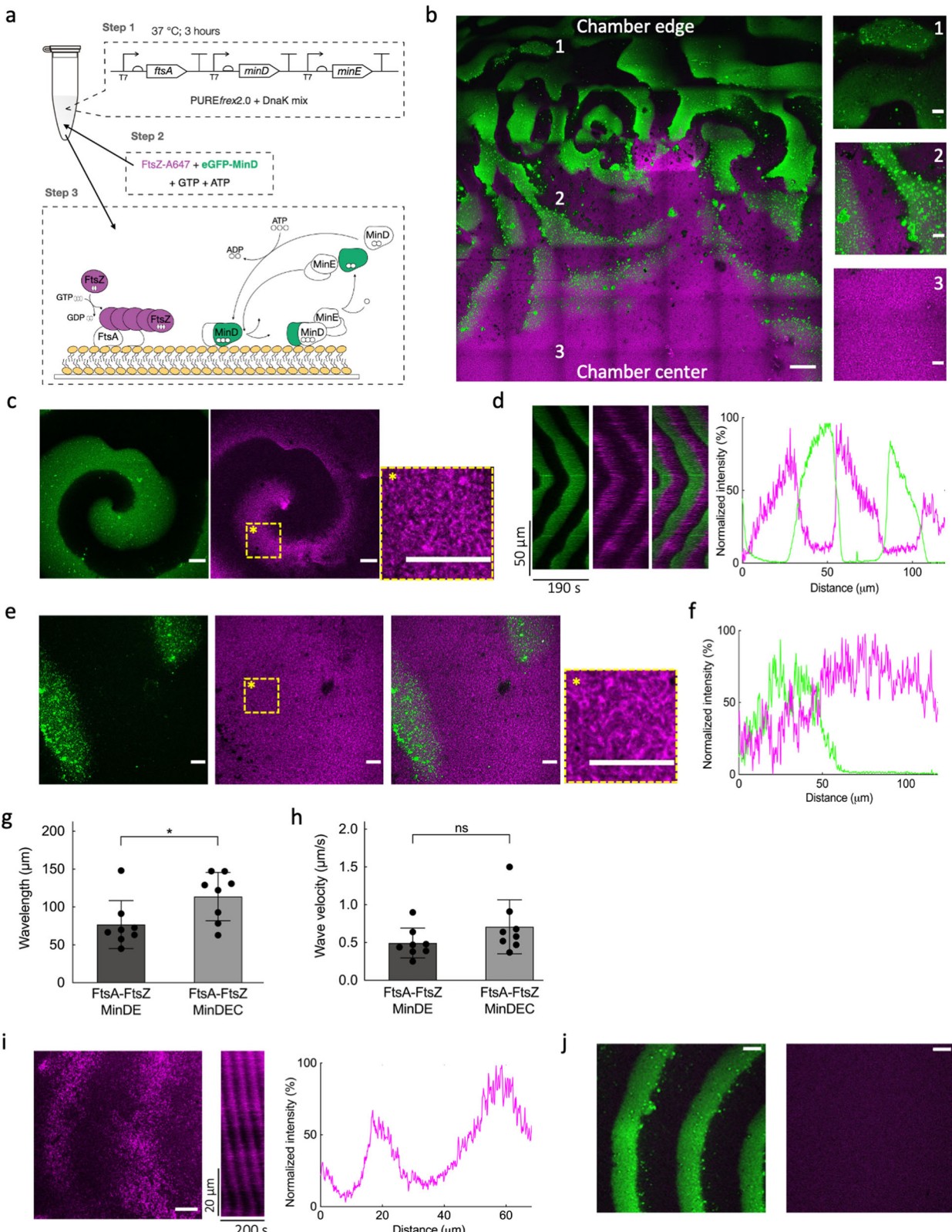

Low-amplitude anticorrelated patterning of FtsA has already been reported with purified proteins in simple buffer conditions[27]. Here, a PURE solution containing pre-expressed FtsA, MinD and MinE was supplemented with purified FtsA conjugated to the AlexaFluor488 fluorophore (0.4 μM) for imaging (Fig. S4c). MinDE waves were able to generate sharp FtsA dynamic patterns

even in the absence of FtsZ. This finding corroborates previous observations with peripheral membrane proteins, but the modulation is noticeably more pronounced in our assay than previously reported[27].

We performed a series of control experiments that confirm that MinDE constitutes the minimal set of proteins to re-organize

**Fig. 3 MinDE proteins, without MinC, regulate FtsA-FtsZ patterns on supported membranes. a** Schematic of the experimental workflow for end-point expression assays. Genes *ftsA*, *minD* and *minE* were co-expressed from a single DNA template (5 nM) in the PURE system in the presence of DnaK chaperone mix for 3 h at 37 °C. The solution was then supplemented with 2.5 mM ATP, 2.0 mM GTP, 100 nM eGFP-MinD and 3 μM FtsZ-A647 before transfer on top of an SLB. **b** Mosaic of 7 × 7 tile scan microscope images showing large-scale organization of FtsA-FtsZ and MinDE dynamic patterns. FtsA-anchored FtsZ filaments mostly populate the center of the chamber (3), while MinDE oscillating gradients dominate at the edges of the chamber (1). In between these two areas (2), correlated patterns of the two subsystems can be seen. Signals from eGFP-MinD and FtsZ-A647 are in green and magenta, respectively. Composite images of overlaid channels are shown. Scale bar is 50 μm in the mosaic image and 10 μm in the three zoomed-in images. **c** Fluorescence microscopy images of FtsA-FtsZ and MinDE dynamic patterns taken from the intermediate SLB area (2) showing sharp propagating waves of FtsA-FtsZ and MinDE on the SLB. A zoom-in image of the framed area in the FtsZ channel is also displayed, showing the organization of FtsA-anchored FtsZ into curved filaments. Scale bars are 10 μm. **d** The time evolution of the spiral wave in (**c**) was analyzed, and kymographs were constructed. On the right, examples of intensity profiles of MinD and FtsZ are shown. Color coding is the same as for microscopy images. **e** As in (**c**) but low-amplitude propagating waves of FtsZ anticorrelating with MinDE patterns are shown. Scale bars are 10 μm. **f** Example of intensity profiles of MinD and FtsZ corresponding to the images in (**e**). Color coding is the same as for microscopy images. Graphs reporting the calculated wavelength (**g**) and velocity (**h**) for MinDEC and MinDE waves, both in the presence of FtsA-FtsZ. Data are from three biological replicates and two to three fields of view have been analyzed per sample. Bar height represents the mean value, and the standard deviation is appended. Symbols are values for individual fields of view aggregated from three biological replicates. Values obtained for different conditions were statistically compared by performing a two-tailed Welch's *t* test. An asterisk indicates *P* value < 0.05, while "ns" denotes a non-significant difference with *P* value > 0.05. **i** Fluorescence microscopy image of FtsA-FtsZ dynamic patterns on an SLB. Purified MinD reporter was omitted to rule out effects from a possible contamination with MinC. Signals from FtsZ-A647 is in magenta. Scale bars are 10 μm. The corresponding kymograph is displayed, as well as the intensity profile of the FtsZ signal along the direction of wave propagation. Color coding is the same as in the microscopy image. **j** Fluorescence microscopy images of FtsZ and MinDE dynamic patterns without expressed FtsA and with purified eGFP-MinD (100 nM). In this condition, FtsZ is not recruited to the membrane (right image). Signals from eGFP-MinD and FtsZ-A647 are in green and magenta, respectively. Scale bars are 10 μm.

FtsA-FtsZ cytoskeletal structures on SLBs (Fig. 3i, j). We excluded the possibility that contaminating amounts of MinC originating from purification carry over in the eGFP-MinD sample could influence the results by performing the same experiment as in Fig. 3b, this time omitting eGFP-MinD and using FtsZ-A647 as the only fluorescent reporter. Similar FtsZ oscillatory gradients were observed (Fig. 3i), confirming our finding that MinDE constitutes the minimal set of proteins to re-organize FtsA-FtsZ cytoskeletal structures on SLBs. In another control, FtsA was omitted by expressing a DNA template encoding only for MinD and MinE. FtsZ was not recruited to the membrane, neither within nor outside the MinDE patterns (Fig. 3j). The measured wavelength was 46 ± 11 μm (Fig. S4b), further indicating that the presence of FtsA-FtsZ on the membrane increases the Min wavelength. When only MinD was expressed, no FtsZ recruitment was observed, ruling out the possibility of MinD-FtsZ interactions (Fig. S4e).

In addition, we demonstrated that in situ protein biogenesis through expression of the three-gene construct on top of the SLB supports the formation of MinDE(C) traveling waves that pattern FtsA-anchored FtsZ filaments (Fig. S5). The two subsystems coexist over the entire membrane surface, which contrasts with the spatial segregation of Min waves and FtsA-FtsZ cytoskeletal structures when pre-expressed proteins were added to the imaging chamber (Figs. 2b and 3b).

**MinC is dispensable for coupling the Min waves and FtsA-FtsZ in closed compartments**. We investigated if the findings from flat bilayer assays were also valid in a closed environment that better mimics the cellular context in terms of membrane surface-to-volume ratio and finite protein numbers. To do so, the FtsA-FtsZ and Min systems were co-reconstituted within lipid microdroplets. We first expressed the tri-cistronic DNA template in a test tube and the pre-ran PUREfrex2.0 solution was encapsulated inside water-in-oil droplets (Fig. 4a). The vast majority of the droplets displayed both FtsZ and Min protein signals on the inner surface (Figs. S6a and S7a, Supplementary Movie 5). Only a few droplets exhibited either Min dynamic patterns or FtsA-FtsZ networks. Time lapse imaging showed that in both samples with (Fig. 4b, Fig. S6 and Supplementary Movie 6) and without MinC (Fig. 4c, Fig. S7 and Supplementary Movie 6), dynamic

concentration gradients of Min and FtsZ formed at the lipid monolayer, and that the two patterns were anticorrelated, as observed on SLBs. In most of the droplets, FtsZ and MinDE(C) assembled on opposite sides of the membrane forming two or multiple domains (Fig. 4b, c, Figs. S6a–c and S7a–c). These patches were not static, they migrated along the droplets surface (Fig. 4b, c, Supplementary Movie 6). Other droplets exhibited circling patterns in which MinDE(C) and FtsZ chased one another along the droplet interface, with no visible intraluminal diffusion within the imaging plane (Fig. 4b and Supplementary Movie 6). In droplets larger than 20 μm in diameter, it was possible to visualize cytoskeletal FtsZ structures forming a cortex and being re-localized by Min waves (Fig. 4d, Figs. S6d and S7d, and Supplementary Movie 7). More complex dynamic patterns were also observed in large droplets, including multiple interfacial FtsZ polarization sites rearranged by the Min dynamics (Fig. 4e, and Figs. S6 and S7), resembling the standing Min waves reported on SLBs. For the different phenotypes the antiphase localization of the two subsystems persisted in time. Parameter values for the pattern dynamics were broad, with wavelength and wave velocity respectively ranging between 42–174 μm and 0.16–0.58 μm s$^{-1}$ (8 droplets) with MinC, and between 24–167 μm and 0.23–0.60 μm s$^{-1}$ (8 droplets) without MinC. These values were consistent with those found on SLB. No differences due to the presence or absence of MinC could be measured.

These data confirm the functional integration of FtsA-FtsZ and Min proteins in closed compartment, further demonstrating that MinC is not essential to create dynamic FtsZ patterns in vitro when FtsA acts as the membrane anchor.

**Effects of the FtsZ membrane anchor protein**. We next sought to explore whether a change in the FtsZ membrane anchor protein might affect the MinDE-driven re-localization of the FtsZ filaments. To do so, FtsA was replaced by either the hypermorphic mutant FtsA(R286W), also known as FtsA*, or ZipA.

Compared to the wild-type protein, FtsA* has a reduced propensity to form oligomers, which improves the lateral interactions between FtsZ protofilaments[19,43]. In vivo, FtsA* promotes the recruitment of division proteins and stabilizes FtsZ ring rearrangement[43–45]. In vitro, purified FtsA* has been shown to assist FtsZ in constricting liposomes[46]. The gene for FtsA* and

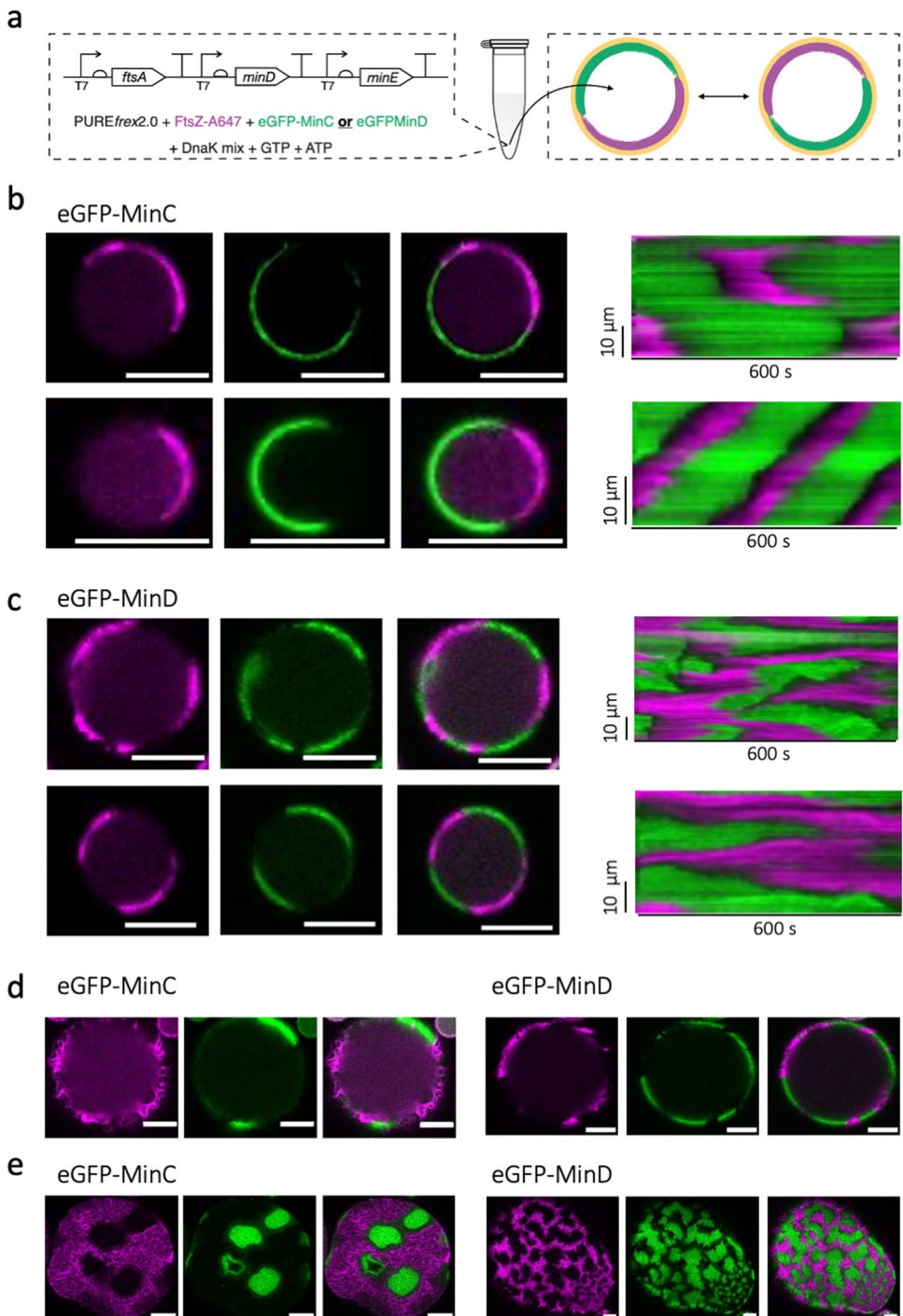

**Fig. 4 Functional reconstitution of MinDE(C) and FtsA-FtsZ networks in microdroplets. a** Schematic illustration of the droplet assays. The DNA template containing the genes *ftsA*, *minD*, and *minE* (5 nM) was expressed in a test tube in the presence of DnaK mix. The pre-ran PURE*frex*2.0 solution was encapsulated inside water-in-oil droplets, along with ATP (2.5 mM), GTP (2 mM), FtsZ-A647 (3 μM) and either purified eGFP-MinC (0.5 μM) or eGFP-MinD (100 nM). **b** Fluorescence images (split channels and composite) of two droplets exhibiting antiphase dynamic patterns of MinDEC and FtsA-FtsZ. The corresponding kymographs are displayed. The droplet at the bottom exhibited a circling pattern yielding a characteristic kymograph. **c** Same as in (**b**), except that eGFP-MinC was substituted with eGFP-MinD. **d** Same as in (**b**) and (**c**) (exact condition is as specified) except that bigger droplets (diameter > 20 μm) were imaged. Here, more defined cytoskeletal FtsZ structures were visible. **e** Same as in (**b**–**d**) (exact condition is as specified) except that the images have been acquired closer to the dome of large droplets. Here, multiple interfacial FtsZ polarization sites rearranged by the Min dynamics were visible. Color coding: eGFP-MinC and eGFP-MinD (green), FtsZ-A647 (magenta). Scale bars in all droplet images are 10 μm.

the *minD-minE* bi-cistron DNA were separately expressed in test-tubes in the presence of DnaK mix. Then, the two reaction solutions were mixed in a 1:1 ratio, supplemented with purified FtsZ-A647 and the other usual reagents, and transferred onto an SLB. Min waves preferentially occupied the bilayer areas at the edge of the chamber, while FtsA*-FtsZ structures were restricted to the center of the chamber. The intermediate region, where both MinDE and FtsA*-FtsZ subsystems coexisted, was found to be more expanded than with FtsA wild-type (Fig. S8a). Dynamic patterns of MinD and membrane-anchored FtsZ clearly counter-oscillated, yet the displacement of FtsZ from the membrane by the MinDE waves was less pronounced than with FtsA (Fig. S8b, c). Moreover, line intensity profiles along the propagation direction were wider for FtsZ than for MinD, the peak of FtsZ localizing at the rear of the MinDE oscillations (Fig. S8b). In the presence of MinC, FtsA*-FtsZ structures were responsive to MinDE but to a lesser extent than with FtsA wild-type, since membrane-bound FtsZ polymers were still visible as the Min waves propagated (Fig. S8d). The wavelength and velocity of the Min oscillations were similar with FtsA and FtsA*, as well as the effect of MinC in increasing the wavelength (Fig. S8e, f). Lastly, the FtsA*-FtsZ and Min systems were co-reconstituted within microdroplets, where anticorrelated dynamic patterns formed at the lipid monolayer (Fig. S8g), further confirming that MinC is not an essential element also when FtsA* is used as a membrane anchor.

We then substituted FtsA with ZipA, an integral membrane protein that, together with FtsA, anchors the FtsZ proto-ring to the inner membrane of *E. coli* cells[8]. The gene for full-length ZipA and *minD-minE* DNA were separately expressed in test-tubes. Addition of the crowding agent Ficoll70 was here necessary to promote the formation of FtsZ bundles. Counter-oscillating ZipA-anchored FtsZ and MinDE dynamic patterns were uniformly distributed across the available membrane (Fig. S9a) and not restricted to specific regions of the chamber as observed with FtsA (Fig. 3b) and FtsA* (Fig. S8a). Compared to the sample with FtsA, ZipA-FtsZ filaments were not fully displaced from the membrane by propagating MinDE waves (Fig. S9b, c). A stronger response to the Min oscillating gradient was restored by the presence of MinC (Fig. S9d). As expected from the presence of the bilayer-spanning domain of ZipA, functional reconstitution inside lipid monolayer-stabilized microdroplets was not possible.

Overall, the results show that the reorganization of membrane-anchored FtsZ by MinDE is a common property when using either FtsA, FtsA* or the full-length ZipA. However, the nature of the FtsZ membrane anchor affects the strength of responsiveness to the Min surface waves.

## Discussion

In this work we reconstituted MinDE(C) dynamics with FtsA-FtsZ cytoskeletal networks in a cell-free environment consisting of the PURE system to enable expression of a three-gene DNA template encoding for MinD, MinE and FtsA (Fig. 1). When compared to the simplistic buffers commonly employed in cell-free assays, PURE system provides an environment that closely resembles the complex molecular content of the bacterial cytoplasm. It is possible that some of the PURE system constituents influence the measured processes, for example the ATPase chaperones or ionic components transiently binding to the membrane.

FtsA is an essential, widely conserved membrane anchor protein, playing a major role in Z-ring formation and downstream protein recruitment[7]. It is known that the nature of the FtsA membrane anchor influences the dynamics of FtsZ assembly in SLB assays[21]. Therefore, the questions of whether and how the

Min system drives the spatial rearrangement of FtsA-bound FtsZ are relevant; yet, they have hitherto not been explored.

We found that membrane-bound FtsA-FtsZ cytoskeletal structures organize into dynamic, anti-phased, patterns driven by the MinDEC system (Fig. 2b, c and Fig. S5a). The traditional explanation for the emergence of FtsZ oscillation gradients is that the membrane-tethered polymers reorganize as a result of the interaction with MinC(D). However, similar dynamic behaviors were observed in the absence of MinC (Fig. 3b, c and Fig. S5b, c), demonstrating that MinDE act as a minimal spatiotemporal regulator of FtsA-anchored FtsZ filaments on SLBs. This conclusion is also valid when the two subsystems are enclosed in water-in-oil droplets (Fig. 4).

When we replaced FtsA by the gain-of-function mutant FtsA*, the membrane-anchored FtsZ was responsive to the MinDE oscillations both on SLBs and in closed compartments, yet the efficacy of the displacement was weaker (Fig. S8). The difference may be explained by the higher stability of the cytoskeletal structures containing FtsA*, the longer residence time of filaments on the membrane making them less subject to redistribution by MinDE. It has already been reported that FtsZ and FtsA* recruit each other more efficiently to the membrane during filament formation[19], and that the greater colocalization is mediated by FtsA* filaments having a higher packing density[19,47]. The finding that MinC did not further encourage the redistribution of FtsA*-anchored FtsZ is in agreement with in vivo data showing that FtsA* imparts resistance to high concentrations of MinC, which ordinarily disassembles the Z-ring[44].

So far, FtsZ redistribution on lipid membranes has not been documented in the absence of MinC when ZipA was used in place of FtsA[39]. Dynamic patterns of ZipA-FtsZ only emerged with MinC traveling on MinDE waves. In this previous study, ZipA was not mobile[39], which might have compromised the rearrangement of ZipA-FtsZ filaments by the MinDE oscillatory gradients. Cell-free expression of ZipA allowed us to reconstitute the full-length, mobile, integral membrane protein on a supported lipid bilayer, and to monitor dynamic ZipA-FtsZ patterning by MinDE. The ability of ZipA (combined with Ficoll70) to cross-link FtsZ filaments into high-order network structures may explain the lower vulnerability to MinDE oscillations. Alternatively, ZipA may partially interact with the glass substrate supporting the bilayer, impeding lateral diffusion.

When the Min system was combined with the fusion protein FtsZ-YFP-MTS on SLBs, two distinct scenarios were reported[27]: dynamic FtsZ-YFP-MTS rings were reorganized by MinDE, while static FtsZ-YFP-MTS networks remained unaffected. In both cases, supplementing MinC stimulated formation of clear FtsZ-YFP-MTS patterns[27]. Moreover, FtsZ-YFP-MTS and Min systems have been coupled inside droplets[48] or polydimethylsiloxane compartments[49], showing that Min waves comprising MinC create dynamic FtsZ surface patterns[48,49]. These earlier results involving ZipA[39] or FtsZ-YFP-MTS[27,48,49] differ from the robust dynamic patterning of FtsA-FtsZ reported here in the absence of MinC, indicating that different membrane targeting proteins may respond differently to Min concentration gradients. This conclusion is further supported by our assays with FtsA* and ZipA.

Based on the present observations, both on open and closed membranes, we propose that MinDE oscillations alone constitute the minimal localization mechanism of the FtsA-FtsZ proto-ring. This assumption is reinforced by the recent discovery that MinDE act as a general propagating diffusion barrier exerting steric pressure that can redistribute membrane proteins in vitro[27,28].

In vivo, functional MinC is undoubtedly needed to prevent septum misplacement and asymmetric division, which results in the production of minicells[11]. The dispensable function of MinC in regulating FtsZ-FtsA dynamics, as reported here, suggests that

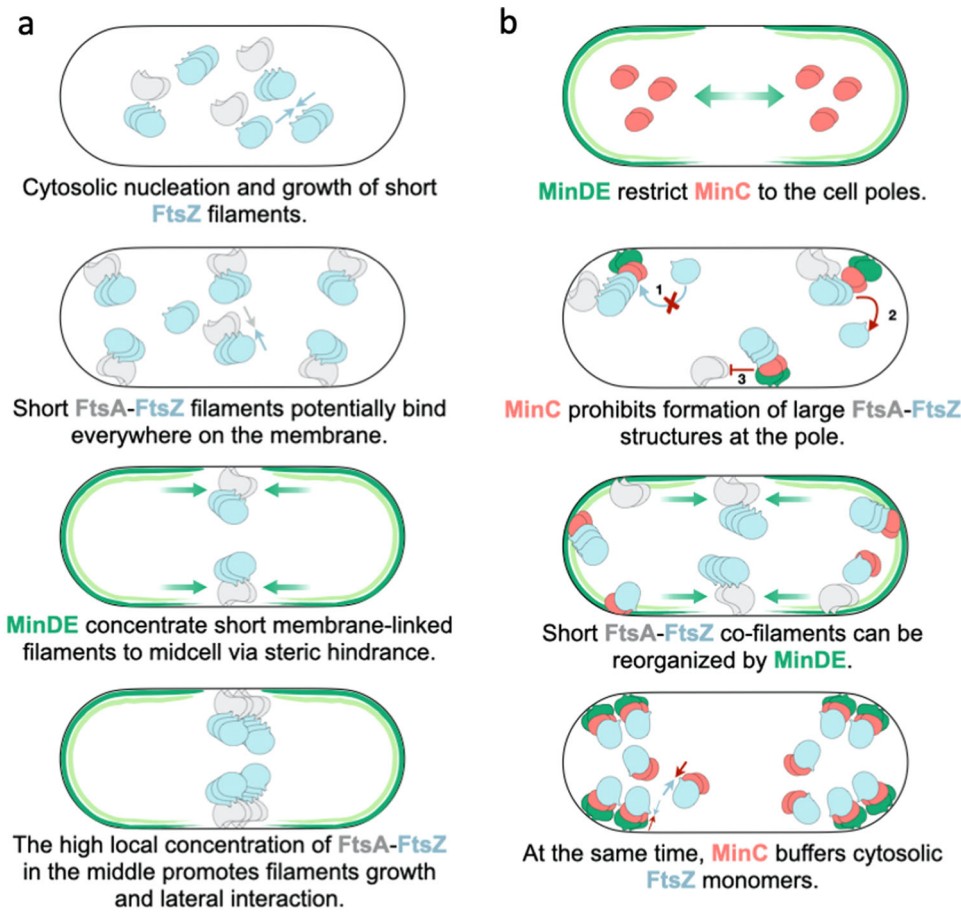

**Fig. 5 Model for the patterning of bacterial cytokinesis components by MinDE oscillations in *E. coli*. a** Short protofilaments of FtsA-FtsZ nucleate and grow in the cytosol. The short FtsZ-FtsA (and possibly ZipA-FtsZ) filaments bind at random locations along the membrane. MinDE propagating diffusion barrier concentrates the short filaments at midcell. The high local concentration of short filaments in the middle promotes lateral interaction, bundling and extension of filaments, and subsequent proto-ring formation. **b** MinD-MinE dynamics generate a concentration maximum of MinC at the poles, where the formation of extended FtsA-FtsZ structures that could not be displaced by MinDE only is prohibited by the following processes: MinCD inhibits long FtsA-FtsZ structures by capping growing FtsZ polymers (1), by depolymerizing FtsZ structures (2), and by competing with other membrane proteins for binding to FtsZ (3). The short co-filaments have a reduced lifetime at the membrane and can more easily be outcompeted by MinDE at the division site. Furthermore, MinC is responsible for sequestering FtsZ monomers through direct interaction, keeping the cytosolic concentration of FtsZ lower than the threshold to form long filaments and bundles. The interaction between membrane-bound MinD and MinC boosts the local concentration of MinC, which might operate as a mechanism to accentuate the depletion of FtsZ subunits. Together, the processes described in (**a**) and (**b**) restrict proto-ring formation to midcell.

MinDE oscillations could play a role in patterning critical division components in *E. coli* than solely distributing MinC at the cell pole, where Z-ring formation is prohibited. After all, in vivo studies have shown that ZipA, as well as other early division proteins, counter-oscillate with MinC[50]. ZipA traveling in phase with FtsZ was interpreted as resulting from the interaction between these two proteins during filament depolymerization by MinC. However, ZipA might also be redistributed by the MinDE oscillations. FtsA and ZipA follow the FtsZ pattern in time, and the three proteins are detectable at midcell with no substantial delay[32], suggesting a tightly coordinated regulatory mechanism. The earliest phase of cytokinesis in *E. coli* includes the cytosolic assembly of short FtsZ protofilaments that bind transiently everywhere on the inner membrane and, over time, migrate and cluster at the cell's center[51]. Our results suggest that MinDE may promote the localization of FtsA-anchored FtsZ complexes at midcell by acting as a propagating steric hindrance on the membrane, competing with the fast detaching-binding short filaments (Fig. 5a). The higher local concentration of FtsA-FtsZ complexes in the middle of the cell would then encourage lateral interaction, bundling and extension of filaments, and possibly proto-ring formation. As shown in Fig. 3e, in some instances FtsA-FtsZ structures cannot effectively be displaced by MinDE only. MinC inhibits long FtsA-anchored FtsZ filaments by depolymerizing membrane-bound FtsZ structures, by competing with the membrane proteins for binding to FtsZ[42], and by capping growing polymers[52]. By lowering the residence time of FtsA-anchored FtsZ filaments on the membrane, MinC may assist MinDE in effectively reorganizing the FtsA-FtsZ structures (Fig. 5b). In addition, MinC could be critical in managing the pool of FtsZ that is not at the division site (up to 70% of the available FtsZ[53]), i.e. by buffering the concentration of cytosolic FtsZ. The observation that FtsZ travels in phase with MinC in the absence of a membrane anchor lends support to this hypothesis (Fig. 2e). In *E. coli*, MinC overexpression impairs cytokinesis, resulting in cell filamentation[54]. The inhibitory effect is caused by MinC sequestering a large pool of free FtsZ[55]. This depletion process is likely enhanced by the binding of MinC to MinD, which increases the local concentration of MinC at the membrane. We acknowledge that many physiological factors, such as

molecular crowding at the cytoplasmic membrane, MinDE wavelength, and protein stoichiometry, might influence the dynamic interplay between the Min and FtsA-FtsZ subsystems, which cannot effectively be reproduced in our cell-free assays. In vivo studies will eventually be necessary to validate this model.

It should be noted that in *E. coli*, the lack of Min system does not totally hinder division; Z-rings can develop both at midcell and at cell poles, culminating in minicells and cells with regular or altered sizes[56,57]. Furthermore, in slow growing *E. coli* cells, condensation of membrane-bound FtsZ filaments into a ring occurs in the absence of the Min system[58], indicating that other mechanisms guide the ring assembly and its localization. For example, nucleoid occlusion and other unknown factors may assist in positioning cell division. SlmA, bound directly to lipid membranes and DNA, could play a role in the spatiotemporal modulation of Z-ring assembly by narrowing the division site at midcell[59].

Finally, our finding will have implications toward the reconstitution of a more elementary regulatory mechanism for positioning the division site in a synthetic cell. DNA-encoded protein synthesis and compartmentalization are important design elements to build a minimal cell. Therefore, demonstrating that the FtsZ and Min subsystems can be partly expressed from a multigene DNA template, functionally integrated, and encapsulated in droplets is a step forward to establishing a synthetic division mechanism. Implementation of a coupled FtsA-FtsZ-MinDE(C) system inside deformable lipid vesicles[25,31] might be a route to symmetrical division of artificial cells. Spatiotemporal redistribution of other bacterial division proteins, such as ZapA, ZapB, MatP, and of lipid synthesis enzymes[60] could also be explored in the future.

## Materials and methods

**Purified proteins**. The proteins eGFP-MinD and eGFP-MinC were purified using previously described methods[24]. Protein concentrations were quantified via Bradford assay and eGFP absorbance measurements. FtsZ was dialyzed against 20 mM Hepes/HCl, pH 8.0, with 50 mM KCl, 5 mM MgCl$_2$, and 1 mM ethylenediaminetetraacetic acid (EDTA). The protein was initially polymerized at 30 °C using 20 mM CaCl$_2$ and 2 mM GTP, then the mixture was incubated at 30 °C for 15 min with a 20-fold excess of Alexa Fluor 647 (A647). This two-step process helps reduce labeling-induced alterations in FtsZ assembly. The precipitate was resuspended on ice in 50 mM Tris/HCl, pH 7.4, with 100 mM KCl, and the unbound fluorescent probe was extracted by gel filtration, as per standard protocol for column purification (Gel Wizard SV). Purified FtsZ-Alexa647 was kept in storage buffer (50 mM Tris, 500 mM KCl, 5 mM MgCl$_2$, and 5% glycerol) at pH 7, as previously described[61]. FtsA was expressed and purified according to published protocols[62]. Cells were suspended in 50 mM Tris-HCl, 1 mM TCEP, 1 mM PMSF, pH 7.5, treated with 10 g/mL DNase, sonicated and centrifuged. The pellet was washed two times with 20 mM Tris-HCl, 10 mM EDTA, pH 7.5, and 1% (v/v) Triton X-100. Inclusion bodies were dissolved in 20 mM Tris-HCl, 5 M guanidine-HCl, 0.5 M NaCl, pH 7.5, loaded in a HisTrap FF column (GE Healthcare) and rinsed in 20 mM imidazole. An imidazole elution gradient (20–500 mM) was applied and the eluted fractions were kept at −80 °C. FtsA was refolded by dialysis against 50 mM Tris-HCl, 0.5 M NaCl, 5 mM MgCl$_2$, 0.2 mM TCEP, and 0.1 mM ADP, pH 8 at 4 °C. Buffer was replaced with 50 mM Tris-HCl, 0.5 M KCl, 5 mM MgCl$_2$, 0.2 mM TCEP and 0.1 mM ADP, pH 7.5. Concentration of FtsA was measured by Bradford assay. FtsA was fluorescently labeled with AlexaFluor-488 (1:10 molar ratio) for 30 min in 50 mM Hepes/HCl (pH 8.0), 100 mM KCl, and 5 mM MgCl$_2$.

**Preparation of DNA constructs**. The plasmid DE was assembled via Gibson assembly (Gibson Assembly Master Mix of New England BioLabs, Inc.). The assembly was performed at equimolar concentrations of linearized plasmid MinDpUC57 and MinE-coding DNA fragments for 1 h at 50 °C. The primers in Table S1 were used: ChD173 + 504 and ChD1139 + 1140. Transformation of the Gibson assembly products into *E. coli* TOP10 competent cells was done by heat shock, after which cells were resuspended in 200 µL of fresh prechilled liquid lysogeny broth (LB) medium and incubated for 1 h at 37 °C and 250 r.p.m. Then, the cultures were plated in solid LB medium with ampicillin and grew overnight at 37 °C. Colonies were picked up and cultured in 1 mL of liquid LB medium with 100 µg mL$^{-1}$ of ampicillin for 16 h at 37 °C and 250 r.p.m. Plasmid purification was performed using the PureYieldTM Plasmid Miniprep System (Promega). Concentration and purity of DNA were checked on NanoDrop. Linear DNA construct

was prepared by polymerase chain reaction (PCR) using the forward and reverse primers ChD365 and ChD174, respectively, annealing to the T7 promoter and T7 terminator sequences (Supplementary Table1).

The plasmid ADE was assembled via Gibson assembly, using equimolar concentrations of linearized plasmid FtsApU57 and the MinDE DNA fragment, for 1 h at 50 °C. The primers in Table S1 were used: ChD420 + 1145 and ChD365 + 174. Transformation of the Gibson assembly products into *E. coli* TOP10 competent cells was done by heat shock, after which cells were resuspended in 200 µL of fresh prechilled liquid lysogeny broth (LB) medium and incubated for 1 h at 37 °C and 250 r.p.m. Then, the cultures were plated in solid LB medium with ampicillin and grew overnight at 37 °C. Colonies were picked up and cultured in 1 mL of liquid LB medium with 100 µg mL$^{-1}$ of ampicillin for 16 h at 37 °C and 250 r.p.m. Plasmid purification was performed using the PureYieldTM Plasmid Miniprep System (Promega). Concentration and purity of DNA were checked on NanoDrop. Linear DNA construct was prepared by PCR using the forward and reverse primers ChD1187 and ChD174, respectively, annealing to the T7 promoter and T7 terminator sequences. For both linear constructs (DE and ADE) amplification products were checked on a 1% agarose gel and were further purified using Wizard SV gel (standard column protocol). Concentration and purity were measured by NanoDrop. All sequences were confirmed by sequencing. MinE-, and FtsA-coding DNA fragments (starting with a T7 promoter and ending with a T7 terminator) were sequence-optimized for codon usage, CG content, and 5′ mRNA secondary structures. Sequences of the linearized constructs can be found in the Supplementary Methods.

**Cell-free gene expression in bulk**. Gene expression was performed using PURE*frex*2.0 (GeneFrontier Corporation, Japan) and 5 nM of a linear DNA template following the supplier's recommendations. The solution was supplemented with 1 µL of DnaK Mix (GeneFrontier Corporation), which consists of purified DnaK (5 µM), DnaJ (1 µM), and GrpE (1 µM) chaperone proteins from *E. coli* (final concentrations are given). Twenty microliter reactions were run in PCR tubes for 3 h at 37 °C. Purified proteins (FtsZ-A647, eGFP-MinC/D), ATP and/or GTP were supplied to the mixture only when specified.

**QconCAT purification**. A quantitative concatemer (QconCAT) protein was designed to contain two specific peptides for FtsA and MinD, one for MinE and two for ribosomal core proteins (Supplementary Table 2). These include the most C-terminal proteolytic peptide that we could accurately identify for each protein of interest[25,31]. QconCAT was expressed in BL21(DE3) cells in M9 medium with $^{15}$NH$_4$Cl and ampicillin. A pre-culture was diluted 1:100 to a 50 mL expression culture. Protein expression was induced at OD$_{600}$ = 0.5 with 1 mM isopropyl β-d-1-thiogalactopyranoside and cells were grown for 3 h at 37 °C. Cells were harvested by centrifugation and the pellet was dissolved in 1 mL B-PER. 10 µL of 10 mg mL$^{-1}$ lysozyme and 10 µL of DNaseI (ThermoScientific, 1 U µL$^{-1}$) were added and the sample was incubated for 10 min at room temperature. The lysate was centrifuged for 20 min at 16,000 × g and the pellet resuspended in 2 mL of a 1:10 dilution of B-PER in MilliQ water. The sample was twice again centrifuged, and the pellet was resuspended in 2 mL 1:10 diluted B-PER and centrifuged again. The pellet was resuspended in 600 µL of 10 mM Tris-HCl pH 8.0, 6 M guanidinium chloride and incubated at room temperature for 30 min. After spinning down the insolubilized protein fraction the supernatant was loaded onto an equilibrated mini NiNTA spin column (Qiagen) and the flow-through was reloaded twice to maximize protein binding. The column was washed twice with 600 µL of 10 mM Tris-HCl pH 6.3, 8 M urea and the QconCAT was eluted with 3 × 200 µL of 10 mM Tris-HCl pH 4.5, 8 M urea, 400 mM imidazole. The eluate was dialyzed overnight and for additional 4 h against 10 mm Tris-HCl pH 8.0, 100 mM KCl with a 10-kDa cutoff slide-a-lyzer cassette (ThermoScientific).

**Trypsin digest**. Per LC-MS injection, 1.5 µL of PURE system reaction was mixed with 3 µL of 100 mM Tris-HCl pH 8.0, 0.3 µL of 20 mM CaCl$_2$, and 0.8 µL MilliQ water. Samples were incubated at 90 °C for 10 min to stop the reaction. Then, 0.6 µL of QconCAT (0.3 mg mL$^{-1}$) was added, the sample was incubated again at 90 °C for 10 min and after cooling to room temperature, 1 µL 250 mM fresh iodoacetamide was added and the sample incubated for 30 min at room temperature in the dark. Then 0.3 µL of 1 mg mL$^{-1}$ trypsin (trypsin-ultra, MS-grade, New England Biolabs) was added and samples were incubated at 37 °C overnight. After addition of 0.7 µL 10% trifluoroacetic acid, samples were centrifuged in a table-top centrifuge (5415 R, Eppendorf) for 10 min at maximum speed. The supernatant was transferred to a glass vial with small-volume insert for LC-MS/MS analysis.

**LC-MS/MS analysis**. LC-MS/MS analysis was performed on a 6460 Triple Quad LCMS system (Agilent Technologies, USA) using Skyline software[63]. Per run 7 µL of sample was injected to an ACQUITY UPLC Peptide CSH C18 Column (Waters Corporation, USA). The peptides were separated in a gradient of buffer A (25 mM formic acid in MilliQ water) and buffer B (50 mM formic acid in acetonitrile) at a flow rate of 500 µL per minute and at a column temperature of 40 °C. The column was equilibrated with 98% buffer A. After injection, the gradient was changed linearly over 20 min to 70% buffer A, over the next 4 min to 60% buffer A, and over

the next 30 s to 20% buffer A. This ratio was held for another 30 s and the column was finally flushed with 98% buffer A to equilibrate for the next run. Selected peptides were measured by multiple reaction monitoring. For both FtsA and MinD, two peptides were analyzed, one for MinE. Two peptides from ribosomal proteins were also measured as a control. The peak area ratio of unlabeled peptides ($^{14}$N, expressed or PURE component) and $^{15}$N-labeled QconCAT peptides was calculated for the ribosomal core proteins (average value over the two peptides) and for the proteins of interest. The concentration of the selected peptides for the proteins of interest was then deduced knowing the concentration of ribosomal core proteins in PURE system. Data processing was performed using a script written in Mathematica (Wolfram Research, version 11.3).

**Phenomenological fitting of FtsA, MinD and MinE production kinetics**. FtsA, MinD, and MinE most C-terminal peptide concentrations were fit using a phenomenological model to estimate apparent kinetic parameters: final yield, production rate, and translation lifetime. The following sigmoid equation was used[64]:

$$y = k' + k \frac{t^n}{t^n + K^n} \tag{1}$$

where $t$ is the time in minutes, $y$ the peptide concentration at a given time point, and $k'$, $k$, $K$, and $n$ are fit parameters. Using this expression, the final yield corresponds to $k$ and the plateau time, or expression lifespan, is expressed as:

$$T_{plateau} = \frac{2K}{n} + K \tag{2}$$

The apparent translation rate, which is defined as the steepness at time $t = K$, is:

$$v_{translation} = \frac{kn}{4K} \tag{3}$$

The kinetics from three independent experiments were fit, and the extracted parameter values are reported as the average and standard deviation.

**Fabrication and cleaning of the imaging chamber**. Home-made imaging glass chambers were used in all the assays[25]. Three microscopy glass slides were bonded together using NOA 61 glue (Norland Products). Holes were drilled across the three-slide layer, with diameters of 2.5 mm. A150-μm thick coverslip (Menzel-Gläser) was attached on one side of the slide with NOA 61 to form the bottom of the chamber. Chambers were cleaned by 10-min washing steps in a bath sonicator (Sonorex Digitec, Bandelin) using the following solutions: chloroform and methanol (volume 1:1), 2% Hellmanex, 1 M KOH, 100% ethanol, and MilliQ water. In addition, after a couple of experiments the glass chambers were subjected to acid Piranha treatment.

**Lipids**. 1,2-dioleoyl-sn-glycero-3-phosphocholine (DOPC), 1,2-dioleoyl-sn-glycero-3phosphoglycerol (DOPG) and 1′,3′-bis[1,2-dioleoyl-sn-glycero-3-phospho]-glycerol (18:1 CL) were purchased from Avanti Polar Lipids as chloroform solutions.

**Preparation of small unilamellar vesicles**. Small unilamellar vesicles (SUVs) were used as precursors for SLB production. DOPC (4 μmol) and DOPG (1 μmol) lipids dissolved in chloroform were mixed in a glass vial. A lipid film was formed on the vial wall by solvent evaporation under a moderate flow of argon and desiccated for 30 min at room temperature. The lipid film was resuspended with 400 μL of SLB buffer (50 mM Tris, 300 mM KCl, 5 mM MgCl$_2$, pH 7.5) and the solution was vortexed for a few minutes. The final lipid concentration was 1.25 mg mL$^{-1}$. A two-step extrusion (each of eleven passages) was performed utilizing an Avanti mini extruder (Avanti Polar Lipids) with 250 μL Hamilton syringes, filters (drain disc 10 mm diameter, Whatman), and a polycarbonate membrane with pore size 0.2 μm (step 1) or 0.03 μm (step 2) (Nuclepore track-etched membrane, Whatman).

**Formation of SLBs**. The imaging chamber was treated with oxygen plasma (Harrick Plasma basic plasma cleaner) for 30 min to activate the glass surface. Immediately after, an SUV solution was added to the sample reservoir at a final lipid concentration of 0.94 mg mL$^{-1}$ together with 3 mM CaCl$_2$. The chamber was sealed with a 20 × 20 mm coverslip by using a double-sided adhesive silicone sheet (Life Technologies). After sample incubation at 37 °C for 30 min, the chamber was carefully opened and the SLB was rinsed six times with SLB buffer.

**Activity assays on supported membranes**. The constructs *ftsA-minD-minE* and *minD-minE* were expressed with PURE*frex*2.0 in test tubes as described above and the solution was supplemented with the following compounds: 2 mM GTP, 2.5 mM ATP, 3 μM purified FtsZ-A647, and either 0.5 μM eGFP-MinC or 100 nM purified eGFP-MinD as specified in the text (all final concentrations) in a total volume of 20 μL. The sample was added on top of an SLB, and the imaging chamber was sealed with a 20 × 20 mm coverslip by using a double-sided adhesive silicone sheet. For assays involving in situ expression, a 20 μL PURE*frex*2.0 mixture containing one of the two DNA constructs was supplemented with 2 mM GTP, 2.5 mM ATP, 3 μM purified

FtsZ-A647, and either 0.5 μM eGFP-MinC or 100 nM purified eGFP-MinD as specified in the text (all final concentrations), and the solution was directly incubated on top of an SLB. 0.4 μM purified FtsA-A488 was used only when specified in the main text. The chamber was closed as indicated above, and the sample was immediately imaged with a time lapse fluorescence microscope for up to 5.5 h at 37 °C. Numerous fields of view were acquired at various time points during the expression period.

For the experiments, where different MinE/MinD protein ratios were tested, the reaction mixture was assembled as described above (DNA construct *ftsA-minD-minE* was expressed with PURE*frex*2.0 in test tubes and the solution was supplemented with 2 mM GTP, 2.5 mM ATP, 3 μM purified FtsZ-A647, and 100 nM purified eGFP-MinD), transferred on an SLB and increasing amounts of purified MinD or MinE were added. For the assays with FtsA*, the DNA templates *ftsA** and *minD-minE* were expressed with PURE*frex*2.0 in test tubes and the solution was supplemented with 2 mM GTP, 2.5 mM ATP, 3 μM purified FtsZ-A647, and either 0.5 μM eGFP-MinC or 100 nM purified eGFP-MinD (all final concentrations) in a total volume of 20 μL. In the experiments with ZipA, expression of the genes *zipA* (sequence-optimized for codon usage, CG content, and 5′ mRNA secondary structures) and *minD-minE* was performed in two separate tubes using PURE*frex*2.0 and 5 nM of linear DNA. The solutions were supplemented with 1 μL of DnaK Mix. Twenty microliter reactions were run in PCR tubes at 37 °C for 3 h (MinDE) and 5 h (ZipA). ZipA reaction solution was first incubated on top of an SLB for 10 min at room temperature. In the meantime, the solution containing expressed MinE was supplemented with purified FtsZ-A647 and eGFP-MinD/C, ATP, GTP, and Ficoll70 to a final concentration of 50 g L$^{-1}$, and added to the SLB that was pre-incubated with ZipA at a 1:1 volume ratio.

**Droplet preparation**. In total, 15 mol% 18:1 CL and 85 mol% DOPC in chloroform were mixed in a glass vial. A lipid film was deposited on the wall of the vial upon solvent evaporation through a gentle flow of nitrogen and was further desiccated for 30 min at room temperature. The lipid film was resuspended with mineral oil (Sigma Aldrich, St. Louis, MO) to a final concentration of 2.5 mg mL$^{-1}$ and the solution was vortexed for a few minutes. The inner solution consisted of a pre-ran PURE system solution supplemented with DnaK, 2 mM GTP, 2.5 mM ATP, 3 μM purified FtsZ-A647, and either 0.5 μM eGFP-MinC or 100 nM purified eGFP-MinD as indicated in the text. Five microliter of inner solution was added to 30 μL of the oil/lipid mixture. The droplet-oil mixture was pipetted on a coverslip and imaged with a fluorescence confocal microscope.

**Confocal microscopy**. Droplets and SLBs were imaged using a Nikon A1R Laser scanning confocal microscope with an SR Apo TIRF 100× oil-immersion objective. FtsZ-A647 and eGFP-MinC/D were excited using the 640-nm and 488-nm laser lines, respectively, and suitable emission filters were used. For image acquisition, the software NIS (Nikon) was utilized, with identical settings for all samples. Throughout imaging, samples were kept on a temperature-controlled stage (Tokai Hit INU) that was held at 37 °C.

**Image analysis**. Fiji[65] or MatLab version R2020b were used for image analysis. Fiji's profile plots tool was used to generate fluorescence intensity profiles. The tool displays a two-dimensional graph of pixel intensities along a line drawn in the direction of the moving wave. The wavelength and velocity of the dynamic patterns on SLBs were determined by producing a kymograph parallel to the direction of the traveling wave. Lines were identified using a linear Hough transform after binarization of the kymograph using Sobel edge detection. The slope of the lines equates to the wave velocity, while the distance between the lines corresponds to the wavelength. The frequency of oscillations of a standing wave was determined by computing the autocorrelation function over time for each pixel and identifying the first peak of the function.

**Statistics and reproducibility**. Reproducibility of the experimental findings was validated by preparing samples on different days and conducting the same experiments at least three times independently. The number of biological replicates for each experiment is stated in the figure captions. The microscopy images in the main text figures are representative of the sample properties as assessed from multiple fields of view acquired in at least three independent repeats. No data was omitted from the analysis. The statistical tests employed, reporting confidence intervals, degrees of freedom, and $P$ values are specified in the figure captions.

**Reporting summary**. Further information on research design is available in the Nature Research Reporting Summary linked to this article.

## Data availability

The authors declare that all data supporting the findings of this study are available within the article and its supplementary files. The mass spectrometry-based proteomics data, the Mathematica script for LC-MS/MS data analysis and the individual data points for the graphs in Fig. 1a, c, Fig. S1c, S4d, and S5d are provided in the "Supplementary Data 1" file. All materials and detailed protocols are available on request from the corresponding author.

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

## Acknowledgements

We thank Ilja Westerlaken for purifying the eGFP-MinD and eGFP-MinC proteins and the group of Petra Schwille for providing the corresponding expression plasmids. We also thank Germán Rivas and Mercedes Jimenez for providing us with the purified FtsZ and FtsA and for discussing the data, and Jaan Männik for fruitful discussions. Microscopy measurements were performed at the Kavli Nanolab Imaging Center Delft. This work was financially supported by the Netherlands Organization for Scientific Research (NWO/OCW) through the "BaSyC—Building a Synthetic Cell" Gravitation grant (024.003.019).

## Author contributions

E.G. and C.D. designed experiments and wrote the paper. E.G. and A.D. performed experiments and analyzed data. All the authors discussed the results.

## Competing interests

The authors declare no competing interests.
