## [Peer Review File · Communications Biology]

Reviewers' comments:

Reviewer #1 (Remarks to the Author):

This paper examines the effect of MinCDE oscillation on FtsZ/FtsA structures formed on a supported membrane layer in vitro. They conclude that MinDE counter oscillates with FtsZ/FtsA and may represent the minimal system to spatially regulate FtsZ. There are a number of issues that raise questions about the physiological relevance but also with the interpretation and conclusion as I will outline below. The major issue is that the authors may only be observing an effect on FtsA with the effect on FtsZ being secondary. Thus, it is not shown that this effect is on FtsZ. In fact the lack of an effect of MinC argues strongly that it is not an effect on FtsZ.

A previous publication showed that MinCDE counter oscillated with FtsZ (Schwille's lab: PNAS paper 2013; ref 45 in the list). However, in that study FtsZ was tethered directly to the membrane with an amphipathic helix rather than a natural tether such as FtsA. Interestingly, in that study MinC was necessary to produce the counter oscillation. In another study from the Schwille lab they explored the effect of MinCDE on FtsZ tethered to the membrane by ZipA. They also saw counter oscillation and in data not shown they said it was MinC dependent (ref 38). So, these authors want to repeat this but using FtsA as a tether for FtsZ to see if the system behaves differently. Also, in a later publication the Schwille lab found that MinDE could cause any membrane anchored protein to counter oscillate in the in vitro set up, although never seen in vivo (ref 26). It appears that MinDE displaces the membrane bound protein driving the counter oscillation. This seems inconsistent with the first result where the membrane tethered FtsZ required MinC. However, the membrane tethered FtsZ can polymerize which may resist the effect of MinDE but still be sensitive to MinCDE, since MinC antagonizes FtsZ polymerization.

An added twist to this study is that some of the proteins are translated in vitro (MinD, MinE and FtsA) and added to the SLB along with labelled FtsZ and labelled MinC. They observe the counter oscillation as expected (Fig. 2C&D). If FtsA was omitted then FtsZ was coincident with MinDE, probably due to weak interaction between FtsZ and MinC (Fig. 2E&F).

Next the authors wondered if the observed counter oscillation of FtsA/FtsZ and MinCDE was really dependent upon MinC, or could MinDE drive the counter oscillation. They observe that MinDE can indeed drive the oscillation. If FtsA is omitted then FtsZ was not recruited to the membrane (Fig. 3; this serves as a control for Fig. 2E&F and demonstrates that MinC is responsible for recruiting FtsZ).

Next, they asked what would happen in the absence of FtsZ; they now follow FtsA. They find that MinDE can spatially affect the distribution of FtsA on the membrane. This raises the question of whether FtsZ is even involved in Fig. 2b or are they just seeing an effect of MinDE on FtsA. In other words, is MinDE acting on FtsA similar to what was reported in ref 26, where MinDE was shown to redistribute proteins tethered to the membrane. If so, this would explain why they do not need to add MinC.

Thus, in the end the suggestion that MinDE may be the minimal oscillator to regulate FtsZ is not demonstrated.

Reviewer #2 (Remarks to the Author):

This manuscript highlights the ability of a MinDE complex to spatially regulate FtsA-FtsZ complexes in vitro using both supported lipid bilayers and water-in-oil droplets. The compelling portion of this manuscript is the ability of MinDE to spatially regulate FtsA-FtsZ complexes in the absence of MinC, which is the major regulator of FtsZ spatially in vivo. Overall, the observations and models made in the

manuscript are interesting and may be useful to understanding these in vitro parameters, however, the data provided is not especially useful -- specifically, the data provided by fluorescence image analysis and experimental parameters that is reported in scales that are not physiologically relevant. It is difficult to rationalize the observed behavior in the context of an E. coli cell. Finally, and most importantly, given the work of the Schwille group in this area (i.e., diffusiophoresis) (Ramm, et al., 2021), it is unsurprising the MinDE would re-localize/displace a membrane associated protein complex, and the specificity for A-Z is unclear and perhaps irrelevant.

Additional Comments:

1. The proposed model in which MinDE restricts MinC to the poles of the cell where it inhibits FtsA-FtsZ structures from assembly is plausible; however, the proposal that short FtsA-FtsZ co-filaments are rearranged by MinDE until the high concentration of FtsA-FtsZ at midcell promotes division is far too speculative, especially considering that the conformations/ stoichiometries/assembly numbers of A-Z complexes are entirely unknown. Residence times and diffusion would likely be impacted by stoichiometries and conformations (FtsA membrane association/dissociation notwithstanding).
2. Incorporation of FtsZ or FtsA mutant proteins (defective for interaction, polymerization, membrane association) as controls or additional experiments would strengthen the conclusions and relevance.
3. Wavelengths were calculated at the scale of hundreds of microns. This is expected to be observational but far too large for the cell length (2-4 μm). How was the wavelength calculated? What does this mean for oscillation/Z-ring formation.
4. Experiments incorporating ZipA would add to how one could address specificity.
5. It is unclear why eGFP-MinC was not added to the tri-cistron in cell-free expression instead of adding purified eGFP-MinC to experiments afterwards. The same could be said for eGFP-MinD.
6. It is unclear why a difference in patterning is observed, seemingly randomly, and how the quantification is made. This issue is particularly prevalent in supplemental figure 4C where it reads, "Same as in B but in this sample the protein network self-organized into spirals and planar waves". What are the quantifiable metrics?
7. Supplemental figure 3D and E should be in main figure as it is the control experiment too support that MinDE constitute the minimal set of proteins to re-organize FtsA-FtsZ cytoskeletal structures on SLBs.
8. The proposed model does not strongly support their findings that MinDE does not affect membrane-bound static MinD-FtsZ complexes. Note that MinD does not bind to FtsZ. A perceived MinD-FtsZ interaction is likely non-specific. This needs support from direct biochemical assays in vitro. Moreover, MinC has binding sites for two regions of FtsZ.
9. It is unclear if protein concentrations in the manuscript are presented as monomers or dimers for certain proteins, which is relevant for the stoichiometry. Moreover, MinD:MinE stoichiometry changes modify oscillation rates. How do changes in this ratio impact displacement.
10. Lines 124-125 state that FtsZ traveled in phase with MinC on the MinDE waves. If FtsZ travels with MinC in the absence of FtsA but is anti-correlated with MinC when FtsA is present, it brings up the question of whether or not MinC and FtsA have competing sites on FtsZ, likely in the C-terminal region. This should be addressed further.
11. Define PDMS.
12. Line 272 reads, "It should be noted than in slow growing.." and instead should read, "It should be noted that in slow growing.."
13. The word "reducing" should be changed to "reduce" on line 305
14. A better cartoon/model of the SLB would be helpful in the figures. The imaging chamber is never shown at all either a photo or a cartoon version. This should be shown as it was custom made and not purchased. Furthermore, on this model, GTP and ATP should be depicted in different ways, so they are not represented by the same symbol
15. Are figures 2A and 3A correct stoichiometrically with oligomeric states? MinD trimers (?) not dimers?
16. It is difficult to know how figure 2B correlates to the chamber or the SLB as there is no good picture/model of either to help the reader.

17. It is unclear what "examples" means on line 683 and why examples are even being used in a main figure.
18. Figure 3G and 3H have tick marks for a right y-axis yet no numbers or axis title is present
19. It is unclear how large the droplets in figure 4B are but a diameter of ">20 um" is noted for the droplets in figure 4D.
20. Scale bars between both sets of figures for figure 4E appear to be different sizes but are suggested to both be 10 um
21. It is unclear why the green from eGFP-MinC in the standing/middle figure in supplemental figure S2A is alternating in and out through time lapses
22. There is no indication as to what is fluorescing in supplemental figure S3D. Figure caption states FtsA-FtsZ but does not specify the fluorophore.
23. Is supplemental table S2 referenced in the text?

Reviewer #3 (Remarks to the Author):

The MinCDE proteins are key spatial regulators of the FtsZ cytokinetic ring in *E. coli*. MinD and MinE drive pole to pole oscillation of the complex; MinC binds to FtsZ and prevents its higher-order assembly at polar regions of the cell, resulting in formation of a ring structure at midcell. This process has been largely recapitulated in purified systems using supported lipid bilayers (SLBs), and movement of MinDE waves along the membrane, even without MinC, has been shown to affect the localization patterns of peripheral membrane proteins in these reconstituted systems.

Here, using supported lipid bilayers (SLBs) and lipid vesicles as cytomimetic environments, the authors extend the studies of Ramm et al. (ref. 23) to show quite convincingly that waves of MinDE proteins, without MinC, are sufficient to spatially regulate the assembly of FtsZ protofilaments when they are attached to their natural binding partner and membrane anchor, FtsA. One notable feature of the present work is the use of an *in vitro* transcription/translation system to co-synthesize Min proteins and FtsA from a single DNA template, thus producing presumably active proteins in physiological quantities. This bypasses the difficulties in purifying active proteins, particularly FtsA, although the authors did need to purify FtsZ and a denatured/renatured version of FtsA in order to fluorescently label them as tracers. The use of FtsA as the membrane anchor for FtsZ also obviates the need to use artificial FtsZ chimeras, which may not act like the native FtsZ-FtsA complexes.

Overall, the manuscript is well written and presented, and the data are consistent with a model in which moving MinD and MinE complexes on the membrane, independently of MinC, can transiently antagonize FtsZ assembly when FtsZ is tethered to the membrane by FtsA. Conceptually, this is not all that different from what Ramm et al. showed and proposed in their 2018 paper in terms of MinDE acting to redistribute membrane proteins, but it is a useful refinement of the model. The lipid droplet data showing mutual exclusion of membrane association are particularly striking.

However, there are a number of places where this manuscript could be improved. The relationship between the *in vitro* studies and *in vivo* behavior and function of the Min system needs to be discussed with greater depth and accuracy, and there are a number of notable omissions from the literature and misstatements that need correcting (see below). Moreover, a direct interaction between MinD and FtsZ has been reported (PMID 28743721). The authors should account for the possibility that MinD directly affects FtsZ assembly via an MinC-independent mechanism that may not involve competition for the membrane.

Major comments:

- 1) The PURE expression system seems to work very well for these experiments. Nevertheless, unlike other *in vitro* studies with membranes that use purified proteins, in the present case there are numerous other proteins in these S30 extracts, such as ribosomal proteins, RNA polymerase, and the

chaperones DnaK, DnaJ, and GrpE. The authors should mention that it is possible that some of these proteins may influence the behavior on membranes that they are measuring. In particular, DnaK and FtsA are both ATPases and share some structural homology.

2) Along these lines, what is the concentration of the chaperones in the final mixture? Are they as concentrated as FtsA, MinD and MinE?

3) The model in Fig. 5 proposes that MinDE proteins may have a major role in focusing the FtsZ/FtsA/ZipA proto-ring in *E. coli* cells independently of the specific FtsZ antagonist MinC. While the *in vitro* evidence supports this idea, the authors have not accounted for all the other proteins normally at the cytoplasmic membrane in an *E. coli* cell. Although the MinDE proteins become transiently concentrated in local areas during their pole to pole oscillation, it is doubtful that at a thousand or two molecules per cell, they constitute the majority of cytoplasmic membrane-associated proteins in any particular area of the cell. The crowded environment at the *E. coli* cytoplasmic membrane is likely very different than the environment on the SLBs, and the MinDE waves would likely have a much lower effect than in the purified SLB system.

4) One key prediction of the model is that an in-frame Δ minC mutant of *E. coli* (one that is not polar on the downstream minD or minE genes) should generate fewer minicells than a Δ minCDE deletion or a deletion of minD only.

5) The authors should stress that the Min system is not required for FtsZ ring formation or even midcell FtsZ ring formation, as minicell strains still divide at midcell (as well as at the poles) because of nucleoid occlusion and other unknown factors. Fig. 5 and the accompanying text should be modified appropriately.

6) Along the same lines, it should be noted in the Discussion and in the description of the model in Fig. 5 that other factors are important for centering the FtsZ ring in *E. coli* (Rrg. 43), including SlmA-mediated nucleoid occlusion. The recent report of SlmA being able to bind to lipid membranes (PMID 32873767) suggests that SlmA on the DNA and the membrane may directly influence FtsZ binding at the membrane and help to focus the FtsZ ring at midcell.

7) In the same vein, the *B. subtilis* nucleoid occlusion factor Noc uses an amphipathic helix to bind to the membrane and sterically hinders FtsZ from assembling into large structures outside of the Noc-delimited zone around midcell (PMID 33531398). This is somewhat analogous to the authors' model for MinDE-mediated FtsZ assembly patterning.

8) Throughout the manuscript, starting on line 71, the authors use the term "FtsA-FtsZ filaments" or "co-filaments (e.g. line 23 in the Abstract, line 169 in text, Fig. S3 legend heading, etc.). However, this term is misleading, as in *E. coli*, there is no evidence that FtsA forms filaments that align with FtsZ filaments, particularly at the initial proto-ring stage, which is implied by the terms "filaments" and "co-filaments". On the contrary, it was reported that purified FtsA forms mini-rings on lipid membranes. These mini-rings act as anchors and guiding tracks to align FtsZ protofilaments prior to condensation of FtsZ protofilaments into more bundled forms (PMID 28695917). FtsA mutants that bypass early steps in division form curved and straight filaments instead of mini-rings (PMID 29995995), but it is not clear if these structures are present during the proto-ring stage or only at later stages. Therefore, to be more accurate, this term should be changed to something like "FtsA-anchored FtsZ filaments" or, as described in the Introduction, as "FtsA-FtsZ cytoskeletal structures".

9) There are several reports of copolymers formed by MinC and MinD, e.g. PMID 25500731, 29610277, although their existence/relevance in cells is controversial. Were copolymers detected in the SLB system? Would the addition of MinC to the MinDE proteins be expected to assemble into copolymers?

10) The original report of FtsZ counter-oscillating with Min was PMID 15242613.

11) Line 292: It should be recalled that Osawa and Erickson showed that FtsZ tethered to the membrane with a hypermorphic mutant of FtsA (FtsA*) could form rings at the center of liposomes and constrict them (PMID 23776220), so the idea of using FtsA as a natural membrane anchor for FtsZ protofilaments in synthetic "cells" is not new.

Minor comments:

1) Lines 100-102: it should be mentioned here that chaperones were also added to the *in vitro*

membrane systems.

2) Line 104, 123: "ring" or "ring-like" structures are not obvious at all in Fig. 2B, only some structures that are above the background fluorescence. These structures are much clearer in Fig. 2C, although my copy is probably not at sufficiently high resolution to ascertain that they are rings or even ring-like—perhaps string-like would be more appropriate. "Ring" implies closed structures, which are not obvious in these images.

3) Page 4: regarding the wavelength measurements: the wavelength and oscillation time probably depends a great deal on the MinE concentration relative to MinD. Was this tested?

4) Line 126-127: Another explanation for the FtsZ-MinC colocalization is that MinC binds to FtsZ directly in two steps (MinCc-FtsZCterm, then MinCn-FtsZ core domain). This is enough of an interaction to expect colocalization in a purified system. It is also consistent with transient localization of oscillating MinC to constricting FtsZ rings in cells (PMID 10540287).

5) Fig. 3C and E: it is not clear what the scale bars represent.

6) Fig. S2 legend: on line 3, change "left" to "bottom".

Rebuttal letter for:

Manuscript: Min waves without MinC can pattern FtsA-anchored FtsZ filaments on model membranes

Reference: COMMSBIO-21-3395

We are thankful to the Referees for finding our work of 'considerable interest' and for their insightful comments, which helped us strengthen the manuscript. The Referee reports are in **black text** and our point-by-point replies are in **blue text**. Changes in the manuscript (main text and Supplementary information) are in **red text**. All figure numbers, references and page numbers referred in the present letter correspond to the numbering in the revised manuscript.

After considering all the comments and making the necessary modifications, we are delighted to submit a revised manuscript that addresses the referees' concerns. In particular, we included:

1. A new experiment clarifying a potential interaction between MinD and FtsZ (Reviewers #2 and 3).
2. New assays incorporating an FtsA mutant (Reviewers #1 and 3) and ZipA (Reviewer #2).
3. New results about the effects of MinDE protein ratio on the oscillation properties and displacement of FtsA-anchored FtsZ filaments.
4. A more extensive discussion about the physiological relevance and limitations of our findings.

Reviewer #1 (Remarks to the Author):

This paper examines the effect of MinCDE oscillation on FtsZ/FtsA structures formed on a supported membrane layer in vitro. They conclude that MinDE counter oscillates with FtsZ/FtsA and may represent the minimal system to spatially regulate FtsZ. There are a number of issues that raise questions about the physiological relevance but also with the interpretation and conclusion as I will outline below. The major issue is that the authors may only be observing an effect on FtsA with the effect on FtsZ being secondary. Thus, it is not shown that this effect is on FtsZ. In fact the lack of an effect of MinC argues strongly that it is not an effect on FtsZ.

Reply: Throughout the manuscript, we have been cautious to discuss only the regulation of FtsA-anchored FtsZ filaments. The direct effect on FtsZ was never stated nor supported. The spatial

regulation of FtsA/FtsZ structures may be attributable to an effect of the Min system on FtsA (FtsA is indeed displaced by MinDE, Fig. S4 C). The main observation that large protein networks interacting with the membrane can also be displaced provides a valuable expansion of the existing model that presents MinDE as a generic spatial cue for membrane proteins.

The fact that various physiological parameters are missing in reconstituted systems is an intrinsic limitation in all *in vitro* investigations. It should however be acknowledged that running experiments in PURE system simulates the *E. coli* cytoplasm more closely than using conventional buffers, making the dynamic behaviors reported here particularly relevant.

A previous publication showed that MinCDE counter oscillated with FtsZ (Schwille's lab: PNAS paper 2013; ref 45 in the list). However, in that study FtsZ was tethered directly to the membrane with an amphipathic helix rather than a natural tether such as FtsA. Interestingly, in that study MinC was necessary to produce the counter oscillation. In another study from the Schwille lab they explored the effect of MinCDE on FtsZ tethered to the membrane by ZipA. They also saw counter oscillation and in data not shown they said it was MinC dependent (ref 38). So, these authors want to repeat this but using FtsA as a tether for FtsZ to see if the system behaves differently. Also, in a later publication the Schwille lab found that MinDE could cause any membrane anchored protein to counter oscillate in the *in vitro* set up, although never seen *in vivo* (ref 26). It appears that MinDE displaces the membrane bound protein driving the counter oscillation. This seems inconsistent with the first result where the membrane tethered FtsZ required MinC. However, the membrane tethered FtsZ can polymerize which may resist the effect of MinDE but still be sensitive to MinCDE, since MinC antagonizes FtsZ polymerization.

Reply: This is a good overview of the current literature, and we agree with the statements. Our main finding is that, although membrane-tethered FtsZ via FtsA can polymerize and organize into larger structures, the system is still responsive to MinDE oscillations.

An added twist to this study is that some of the proteins are translated *in vitro* (MinD, MinE and FtsA) and added to the SLB along with labelled FtsZ and labelled MinC. They observe the counter oscillation as expected (Fig. 2C&D). If FtsA was omitted then FtsZ was coincident with MinDE, probably due to weak interaction between FtsZ and MinC (Fig. 2E&F).

Reply: This is correct.

Next the authors wondered if the observed counter oscillation of FtsA/FtsZ and MinCDE was really dependent upon MinC, or could MinDE drive the counter oscillation. They observe that MinDE can

indeed drive the oscillation. If FtsA is omitted then FtsZ was not recruited to the membrane (Fig. 3; this serves as a control for Fig. 2E&F and demonstrates that MinC is responsible for recruiting FtsZ). Next, they asked what would happen in the absence of FtsZ; they now follow FtsA. They find that MinDE can spatially affect the distribution of FtsA on the membrane. This raises the question of whether FtsZ is even involved in Fig. 2b or are they just seeing an effect of MinDE on FtsA. In other words, is MinDE acting on FtsA similar to what was reported in ref 26, where MinDE was shown to redistribute proteins tethered to the membrane. If so, this would explain why they do not need to add MinC. Thus, in the end the suggestion that MinDE may be the minimal oscillator to regulate FtsZ is not demonstrated.

Reply: Thank you for providing a comprehensive description of our major results. To the question “is MinDE acting on FtsA similar to what was reported in ref 26 (new ref 27)?”: on lines 177-179, we wrote “*Low-amplitude anticorrelated patterning of FtsA has already been reported with purified proteins in simple buffer conditions*²⁷”. This contrasts to the sharp FtsA dynamic patterns observed here. Although the mechanism driving these two behaviors must be similar, the exact protein concentrations and buffer composition might lead to different responses of FtsA to MinDE waves.

About the last comment, our claim is that MinDE may be the minimal system to regulate large-scale cytoskeletal structures of FtsA-anchored FtsZ; FtsZ alone was never discussed. The additional experiments we performed with the hypermorphic mutant FtsA* (new Fig. S8) do not really help discriminate between a direct or indirect effect on FtsZ because FtsA*, besides a lower propensity to form oligomers than the wild-type protein, increases the lateral interactions between FtsZ filaments, which may also lead to different responses to MinDE oscillations. Of note, under our experimental conditions, with both in-tube and in situ expressed proteins, FtsA most likely binds to the membrane once complexed with short FtsZ filaments, as opposed to a scenario where FtsA binds to the membrane prior interacting with FtsZ, further supporting the idea that MinDE regulates membrane-bound FtsA-FtsZ complexes instead of FtsA alone or FtsZ alone.

Reviewer #2 (Remarks to the Author):

This manuscript highlights the ability of a MinDE complex to spatially regulate FtsA-FtsZ complexes in vitro using both supported lipid bilayers and water-in-oil droplets. The compelling portion of this manuscript is the ability of MinDE to spatially regulated FtsA-FtsZ complexes in the absence of MinC, which the major regulator of FtsZ spatially in vivo. Overall, the observations and models made in the manuscript are interesting and may be useful to understanding thee in vitro parameters, however, the data provided is not especially useful -- specifically, the data provided by fluorescence image

analysis and experimental parameters that is reported in scales that are not physiologically relevant. It is difficult to rationalize the observed behavior in the context of an *E. coli* cell.

Reply: We appreciate the referee's constructive inputs; thank you for finding our observations and model interesting. We agree that the present data are mostly relevant to understanding Min protein dynamics and its interaction with FtsA-FtsZ complexes *in vitro*. Nonetheless, we want to spark a scientific discussion with the *in vivo* bacterial division community too, as we think our findings can help in unveiling some aspects of the spatial control of division proteins in *E. coli*. The discrepancy between the wavelength of Min oscillations *in vitro* and the size of an *E. coli* cell has long been recognized as an intriguing observation. A recent study (PMID: 34083526) showed that the protein bulk concentration gradient orthogonal to the membrane controls Min protein dynamics, bridging *in vitro* and *in vivo* experimental observations in one theoretical framework. Moreover, mechanistic insights revealed in *in vitro* assays have already contributed to better understand Min wave propagation in *E. coli* cells, despite the striking quantitative differences in these two settings. Therefore, we do not believe that the difference of scaling of the Min waves between the here reported model membranes and the bacterial cell precludes any physiological relevance.

Finally, and most importantly, given the work of the Schwille group in this area (i.e., diffusiophoresis) (Ramm, et al., 2021), it is unsurprising the MinDE would re-localize/displace a membrane associated protein complex, and the specificity for A-Z is unclear and perhaps irrelevant.

Reply: We agree that MinDE oscillating gradients may act as a general mechanism to re-localize membrane associated proteins. We have cited the paper of Ramm et al. 2021 and discussed our results in the light of this previous work. Hence, we do not claim that the observed phenomenon is specific to FtsA-FtsZ complexes. However, it remained to be demonstrated that such cytoskeletal structures could be responsive to MinDE waves. Given the modest re-distribution of FtsA reported in Fig. S15 of Ramm et al. 2021, we have been surprised to observe such a striking spatiotemporal regulation of FtsA-FtsZ structures (Fig. 3).

Additional Comments:

1. The proposed model in which MinDE restricts MinC to the poles of the cell where it inhibits FtsA-FtsZ structures from assembly is plausible; however, the proposal that short FtsA-FtsZ co-filaments are rearranged by MinDE until the high concentration of FtsA-FtsZ at midcell promotes division is far too speculative, especially considering that the conformations/ stoichiometries/assembly numbers of

A-Z complexes are entirely unknown. Residence times and diffusion would likely be impacted by stoichiometries and conformations (FtsA membrane association/dissociation notwithstanding).

Reply: We agree that many physiological factors that might influence the dynamic interplay between the Min and FtsA-FtsZ subsystems cannot effectively be reproduced in our assays; this is an inherent limitation of all in vitro studies. Please note, however, that conducting experiments in PURE system offers a closer cytosol-like environment than in any previous studies from other groups.

The hypothesis that short FtsA-FtsZ co-filaments can be rearranged by MinDE and enriched at midcell, promoting proto-ring formation, is reasonable in view of our data. We rephrased the corresponding sentences in the Discussion to temperate the statements and make them read more as a hypothetical scenario. The adjusted text starts from line 338, and reads: *Our results suggest that MinDE may promote the localization of FtsA-anchored FtsZ filaments at midcell by acting as a propagating steric hindrance on the membrane, competing with the fast detaching-binding short filaments (Fig. 5 A). The higher local concentration of FtsA-FtsZ complexes in the middle of the cell would then encourage lateral interaction, bundling and extension of filaments, and possibly proto-ring formation.*

In Ramm et al. 2021, the modest re-localization of FtsA by MinDE led the authors to suggest that “FtsA is counter-oscillating to MinDE in vivo, and depending on the oligomerization state, would possibly be enriched at midcell.” This is in our opinion a sound hypothesis, despite the gap between the in vitro and in vivo settings. Considering that FtsA most likely binds to the membrane in complexed form with FtsZ filaments, our data expands the proposed scenario of Ramm et al. to a more physiological situation encompassing the presence of FtsZ and relevant cytosolic compounds in PURE system.

2. Incorporation of FtsZ or FtsA mutant proteins (defective for interaction, polymerization, membrane association) as controls or additional experiments would strengthen the conclusions and relevance.

Reply: We have already performed controls in which FtsZ was not linked to the membrane and we investigated FtsZ and FtsA in the absence of one another. We agree with the reviewer that additional experiments with an FtsA mutant would strengthen the conclusions. Hence, we cell-free expressed FtsA*, a hypermorphic mutant that has a higher propensity to be monomeric while improving lateral contacts between FtsZ protofilaments compared to the wild-type protein (PMID: 28695917). The new results are discussed on page 7 from line 231.

3. Wavelengths were calculated at the scale of hundreds of microns. This is expected to be observational but far too large for the cell length (2-4 um). How was the wavelength calculated? What does this mean for oscillation/Z-ring formation.

Reply: This in vitro vs in vivo disparity has long been noted: PMID: 27885986, PMID: 26884160, and PMID: 34083526, for example. A recent study (PMID: 34083526) showed that the protein bulk

concentration gradient orthogonal to the membrane controls Min protein dynamics, bridging in vitro and in vivo experimental observations in one theoretical framework. Since no cytosolic components or physiological factors other than the size of the membrane-bound compartment may affect the protein diffusion modes governing the wave properties, we think that our finding that FtsA-anchored FtsZ filaments counter-oscillate MinDE also applies in the cellular context.

The wavelength was calculated as described in the method section on page 17 from line 582.

4. Experiments incorporating ZipA would add to how one could address specificity.

Reply: The Schwille group has explored MinDE dynamics with a range of different membrane proteins and membrane-binding nucleic acids, and the spatiotemporal regulation was described as a nonspecific mechanism (ref 27). We initially did not include ZipA in our experiments because it was shown to be immobile when reconstituted on SLBs and does not respond to MinDE waves (ref 39, although the data were not shown). Prompted by the reviewer's comment and the lack of actual data supporting the claim in ref. 39, we decided to examine the effect of MinDE on the full-length ZipA in our PURE expression reactions. The new results are discussed on page 8 from line 256.

5. It is unclear why eGFP-MinC was not added to the tri-cistron in cell-free expression instead of adding purified eGFP-MinC to experiments afterwards. The same could be said for eGFP-MinD.

Reply: Using purified reporter proteins allows us to work with well-defined concentrations, which is more challenging when expressing proteins, especially when small amounts are needed. It is known that MinC can hinder pattern formation at high concentrations (PMID: 25271375) and impact wavelength and velocity of Min dynamic patterns (PMID: 21516096). Therefore, we aimed at incorporating precise amounts of eGFP-MinC to ensure functionality and physiological relevance. Another reason was to not overload the gene expression machinery as the introduction of an extra gene may create a burden, reducing the amounts of synthesized MinDE/FtsA. In the case of MinD, eGFP-MinD is typically used in conjunction with unlabeled MinD because the fusion protein changes Min dynamics when present in large molar fractions (PMID: 27465495). Mindful of this constraint, we only employed eGFP-MinD at very low concentration, which is difficult to achieve in an accurate manner when starting from the gene. Summing up, using purified eGFP-MinC or eGFP-MinD provides more control and reproducibility, while allowing MinDE and FtsA to be expressed from the tri-cistron DNA at identical levels in all conditions.

6. It is unclear why a difference in patterning is observed, seemingly randomly, and how the quantification is made. This issue is particularly prevalent in supplemental figure 4C where it reads,

“Same as in B but in this sample the protein network self-organized into spirals and planar waves”.

What are the quantifiable metrics?

Reply: The occurrence of different oscillation patterns on supported lipid bilayers is common, and they may be classified as amoebas, mushrooms, bursts, standing waves, traveling waves, and spiral waves based on their appearance (visual inspection of the overall morphology). Different patterns can coexist on the same bilayer, and transitions have also been reported. Among the possible causes of this diversity, differences in the SLB quality, geometry of the chamber, bulk-surface ratio, protein densities, and local concentrations have been invoked. A recent study demonstrated how the bulk-surface ratio influences pattern creation (PMID: 34083526).

In our assays, we kept the conditions constant across all experiments: protocol for SLB formation, fabrication and cleaning of the chambers, expression conditions, etc. However, slightly inhomogeneous washing of the SLB, different degrees of cleaning of the glass support, the overall quality of the small unilamellar vesicles (SUVs) used, and environmental conditions, can constitute factors that affect the membrane quality, resulting in the formation of membrane gaps or minor defects, as well as areas with different lipid mobility. Local differences of the supported lipid bilayer might then cause different patterns within the same chamber. Furthermore, while protein expression is relatively robust, the exact concentration of synthesized proteins might fluctuate from one reaction to another, thereby influencing the pattern dynamics.

The quantification method of the waves is determined by the class of patterns. As outlined in the method section, for standing waves we can quantify the frequency of the oscillations by computing the autocorrelation function over time for each pixel and extracting the first peak of the function. For travelling waves, we can measure wavelength and velocity by generating a kymograph along a line in the direction of the moving wave.

7. Supplemental figure 3D and E should be in main figure as it is the control experiment too support that MinDE constitute the minimal set of proteins to re-organize FtsA-FtsZ cytoskeletal structures on SLBs.

Reply: We agree, the old Fig. S3 D and E now become Fig. 3 I and J. The text has been modified accordingly.

8. The proposed model does not strongly support their findings that MinDE does not affect membrane-bound static MinD-FtsZ complexes. Note that MinD does not bind to FtsZ. A perceived MinD-FtsZ interaction is likely non-specific. This needs support from direct biochemical assays in vitro. Moreover, MinC has binding sites for two regions of FtsZ.

Reply: In our manuscript, we never mentioned membrane-bound MinD-FtsZ complexes. In the control experiment, where FtsA was omitted (Fig. 3 J), FtsZ was not recruited to the membrane despite the presence of MinD. Since this assay was performed in the presence of MinE, which could interfere with a possible MinD-FtsZ interaction, we carried out an additional control (new Fig. S4 E) in which only MinD was expressed. Here too, FtsZ was not recruited to the membrane, further indicating that FtsZ and MinD do not form membrane-bound complexes.

MinC can indeed bind to FtsZ. We observed colocalization of the MinC and FtsZ signals in the absence of FtsA (Fig. 2 E). To emphasize the well-established nature of this interaction, we cited the original paper from the group of Lutkenhaus (new ref. 41) reporting transient localization of the oscillating MinC to the FtsZ ring in cells: *“The observed colocalization of the MinC and FtsZ patterns was consistent with previous results that indicated weak transient interaction between MinC and FtsZ^{40,41}.”*

9. It is unclear if protein concentrations in the manuscript are presented as monomers or dimers for certain proteins, which is relevant for the stoichiometry. Moreover, MinD:MinE stoichiometry changes modify oscillation rates. How do changes in this ratio impact displacement.

Reply: Protein concentrations in the manuscript are presented as monomers, for both purified and expressed proteins.

Mass spectrometry data suggest that the MinE/MinD ratio ranges from about 0 to 1.6 during a PURE reaction (Fig. 1 C). We studied Min dynamics over time in combination with gene expression (Fig. S5) and calculated the oscillation properties at different time points for which the MinE/MinD ratio could be estimated. We found that the MinDE wavelength and velocity are not influenced (new Fig. S5 D) and that the displacement of the membrane-bound FtsA-FtsZ complex is not impacted once the MinDE waves have developed (Fig. S5 A-C). To expand the range of MinE/MinD ratios, while increasing the absolute amounts of proteins, we repeated the assay described in Fig. 3 A, this time by supplementing the reaction solution on the SLB with known quantities of purified MinD or MinE. Changing the MinE/MinD ratio did not influence the wave properties (Fig. S4 D).

The absence of changes in MinDE wave properties and displacement of FtsA-anchored FtsZ filaments is consistent with data reported in ref. 27, where different MinD/MinE ratios (ranging from 10 – 0.1) were shown to not influence pattern dynamics.

10. Lines 124-125 state that FtsZ traveled in phase with MinC on the MinDE waves. If FtsZ travels with MinC in the absence of FtsA but is anti-correlated with MinC when FtsA is present, it brings up the question of whether or not MinC and FtsA have competing sites on FtsZ, likely in the C-terminal region. This should be addressed further.

Reply: A competition between MinC (bound to MinD), FtsA and ZipA for interacting with FtsZ has been reported in vivo (PMID: 19415799). The inhibitory action of MinC requires the C-terminal peptide region of FtsZ that is also involved in the interaction with FtsA and ZipA. Such a competition disrupts the Z-ring by displacing FtsZ polymers from the membrane. We clarified this point by adding the following sentence on lines 129-132: *“The observation that FtsZ travels with MinC in the absence of FtsA, while being anti-correlated with MinC when FtsA is present, is in agreement with a competition mechanism between MinD-bound MinC and the membrane anchoring protein for binding to the C-terminal peptide region of FtsZ⁴².”*

11. Define PDMS.

Reply: Done (line 316).

12. Line 272 reads, “It should be noted than in slow growing..” and instead should read, “It should be noted that in slow growing..”

Reply: Corrected.

13. The word “reducing” should be changed to “reduce” on line 305

Reply: Changed.

14. A better cartoon/model of the SLB would be helpful in the figures. The imaging chamber is never shown at all either a photo or a cartoon version. This should be shown as it was custom made and not purchased.

Reply: We included a new Fig. S2 showing a cartoon and a photo of the chamber.

Furthermore, on this model, GTP and ATP should be depicted in different ways ,so they are not represented by the same symbol

Reply: GTP is now represented by ovals and ATP by circles.

15. Are figures 2A and 3A correct stoichiometrically with oligomeric states? MinD trimers (?) not dimers?

Reply: It was not our intention to display in the cartoon the correct stoichiometry. However, to avoid confusion, we now represent MinD as membrane-bound dimers in both figures.

16. It is difficult to know how figure 2B correlates to the chamber or the SLB as there is no good picture/model of either to help the reader.

Reply: In the new Fig. S2, we included a schematic that shows the location of the three areas (1, 2 and 3) in the chamber.

17. It is unclear what “examples” means on line 683 and why examples are even being used in a main figure.

Reply: We replaced “examples” with “representative”.

18. Figure 3G and 3H have tick marks for a right y-axis yet no numbers or axis title is present

Reply: We changed this style in all the graphs in the manuscript. We removed the top x-axis and the right y-axis from all the graphs.

19. It is unclear how large the droplets in figure 4B are but a diameter of “>20 μm ” is noted for the droplets in figure 4D.

Reply: For clarity, we added: “Scale bars in all droplet images are 10 μm ”.

20. Scale bars between both sets of figures for figure 4E appear to be different sizes but are suggested to both be 10 μm

Reply: The scale bar in all droplet images is always 10 μm , as indicated. We adjusted the dimensions of the images, hence the size of the scale bars, to obtain a well-organized figure.

21. It is unclear why the green from eGFP-MinC in the standing/middle figure in supplemental figure S2A is alternating in and out through time lapses

Reply: This Min pattern is commonly referred as ‘standing wave’ and has already been observed on SLBs (PMID: 26884160 and PMID: 34083526).

22. There is no indication as to what is fluorescing in supplemental figure S3D. Figure caption states FtsA-FtsZ but does not specify the fluorophore.

Reply: We indicated the color coding at the end of the figure caption.

23. Is supplemental table S2 referenced in the text?

Reply: It was not, it is now referenced, thank you.

Reviewer #3 (Remarks to the Author):

The MinCDE proteins are key spatial regulators of the FtsZ cytoskeletal ring in *E. coli*. MinD and MinE drive pole to pole oscillation of the complex; MinC binds to FtsZ and prevents its higher-order assembly at polar regions of the cell, resulting in formation of a ring structure at midcell. This process has been largely recapitulated in purified systems using supported lipid bilayers (SLBs), and movement of MinDE waves along the membrane, even without MinC, has been shown to affect the localization patterns of peripheral membrane proteins in these reconstituted systems.

Here, using supported lipid bilayers (SLBs) and lipid vesicles as cytomimetic environments, the authors extend the studies of Ramm et al. (ref. 23) to show quite convincingly that waves of MinDE proteins, without MinC, are sufficient to spatially regulate the assembly of FtsZ protofilaments when they are attached to their natural binding partner and membrane anchor, FtsA. One notable feature of the present work is the use of an *in vitro* transcription/translation system to co-synthesize Min proteins and FtsA from a single DNA template, thus producing presumably active proteins in physiological quantities. This bypasses the difficulties in purifying active proteins, particularly FtsA, although the authors did need to purify FtsZ and a denatured/renatured version of FtsA in order to fluorescently label them as tracers. The use of FtsA as the membrane anchor for FtsZ also obviates the need to use artificial FtsZ chimeras, which may not act like the native FtsZ-FtsA complexes.

Overall, the manuscript is well written and presented, and the data are consistent with a model in which moving MinD and MinE complexes on the membrane, independently of MinC, can transiently antagonize FtsZ assembly when FtsZ is tethered to the membrane by FtsA. Conceptually, this is not all that different from what Ramm et al. showed and proposed in their 2018 paper in terms of MinDE acting to redistribute membrane proteins, but it is a useful refinement of the model. The lipid droplet data showing mutual exclusion of membrane association are particularly striking.

Reply: We thank the referee for the positive comments.

However, there are a number of places where this manuscript could be improved. The relationship between the *in vitro* studies and *in vivo* behavior and function of the Min system needs to be discussed with greater depth and accuracy, and there are a number of notable omissions from the literature and misstatements that need correcting (see below). Moreover, a direct interaction between MinD and FtsZ has been reported (PMID 28743721). The authors should account for the possibility that MinD directly affects FtsZ assembly via an MinC-independent mechanism that may not involve competition for the membrane.

Reply: Thank you for the constructive comments. We addressed them below.

Direct interaction between MinD and FtsZ has been described in the absence of MinE (PMID 28743721). In Fig. 3 J, where FtsA was omitted, FtsZ was not recruited to the MinDE waves. Moreover, we did not find evidence for direct MinD-FtsZ interaction at time zero in Fig. S5 B and C, where both eGFP-MinD (traces) and FtsZ were present. To further investigate the possibility of an interaction, we performed an assay with cell-free expressed MinD and purified FtsZ (without MinE). No membrane recruitment of FtsZ nor colocalization of the two proteins could be observed (new Fig. S4 E). This result was mentioned in lines 194-195 of the revised manuscript. The discrepancy with the previous work might be explained by fact that the MinD-FtsZ interaction was explored in PMID 28743721 in the absence of a membrane. It is well understood that MinD attachment to the membrane influences its interactions with other proteins; for example, MinD mutants that are unable to adhere to the membrane interact less with MinC (PMID: 14973039). This might also explain why this interaction has never been found in earlier investigations on supported lipid bilayers.

Major comments:

1) The PURE expression system seems to work very well for these experiments. Nevertheless, unlike other in vitro studies with membranes that use purified proteins, in the present case there are numerous other proteins in these S30 extracts, such as ribosomal proteins, RNA polymerase, and the chaperones DnaK, DnaJ, and GrpE. The authors should mention that it is possible that some of these proteins may influence the behavior on membranes that they are measuring. In particular, DnaK and FtsA are both ATPases and share some structural homology.

Reply: The PURE system used in this manuscript is not an S30 extract, but a reconstituted *E. coli*-based transcription-translation system. It contains highly purified components for gene expression and was supplemented with the chaperones DnaK, DnaJ, and GrpE for functional biosynthesis of FtsA and MinE. However, one should acknowledge that using PURE system provides an environment that more closely resembles the *E. coli* cytoplasm than in cell-free assays performed in oversimplified buffers. Therefore, we expect that the observed behaviors are more physiologically relevant, albeit quantitatively different than in vivo.

As suggested, we added the following sentences starting on line 276: *“When compared to the simplistic buffers commonly employed in cell-free assays, PURE system provides an environment that closely resembles the complex molecular content of the bacterial cytoplasm. It is possible that some of the PURE system constituents influence the measured processes, for example the ATPase chaperones or ionic components transiently binding to the membrane.”*

2) Along these lines, what is the concentration of the chaperones in the final mixture? Are they as concentrated as FtsA, MinD and MinE?

Reply: The concentration of the chaperones in the final mixture are 5 μM DnaK, 1 μM DnaJ and 1 μM GrpE. We added this information in the Methods section, line 439. The chaperones are indeed in the concentration range of the expressed proteins.

3) The model in Fig. 5 proposes that MinDE proteins may have a major role in focusing the FtsZ/FtsA/ZipA proto-ring in *E. coli* cells independently of the specific FtsZ antagonist MinC. While the in vitro evidence supports this idea, the authors have not accounted for all the other proteins normally at the cytoplasmic membrane in an *E. coli* cell. Although the MinDE proteins become transiently concentrated in local areas during their pole to pole oscillation, it is doubtful that at a thousand or two molecules per cell, they constitute the majority of cytoplasmic membrane-associated proteins in any particular area of the cell. The crowded environment at the *E. coli* cytoplasmic membrane is likely very different than the environment on the SLBs, and the MinDE waves would likely have a much lower effect than in the purified SLB system.

Reply: We generally agree with the reviewer's comment that the properties of the *E. coli* cytoplasmic membrane, in particular molecular crowding, most likely differ from those on an SLB, with possible consequences on the effects of MinDE waves in these two environments. Nonetheless, results from in vivo studies show that the MinCDE system is involved in membrane protein re-localization despite the rather low abundance of MinD and MinE proteins. For instance, (i) a proteomic analysis showed that the presence of MinCDE reduces the abundance of several membrane-bound proteins (ref 26), (ii) incorrect localization of some membrane proteins was observed in the absence of MinCDE (PMID: 22380631), and (iii) ZipA was found to counter-oscillate MinCDE (ref. 50).

Experiments regarding protein density were not included, since this issue was largely experimentally addressed on SLBs in ref. 27. The authors demonstrated that the placement of membrane-anchored proteins can robustly be seen for MinD/MinE ratios ranging from 10 to 0.1. The antiphase pattern was observed at membrane protein/MinDE ratios ranging from 30 to 0.1. MinDE was also able to pattern a considerably crowded membrane constituted of lipid-anchored streptavidin (6600 molecules/ μm^2). We acknowledge that many physiological factors might influence the dynamic interplay between the Min and FtsA-FtsZ subsystems, which cannot effectively be reproduced in our assays; this is an inherent limitation of all in vitro studies. Please note, however, that conducting experiments in PURE system offers a closer cytosol-like environment than in any previous studies from other groups.

We added the following sentence in the main text, lines 353-357: *“We acknowledge that many physiological factors, such as molecular crowding at the cytoplasmic membrane, MinDE wavelength, and protein stoichiometry, might influence the dynamic interplay between the Min and FtsA-FtsZ subsystems, which cannot effectively be reproduced in our cell-free assays. In vivo studies will eventually be necessary to validate this model.”*

4) One key prediction of the model is that an in-frame Δ minC mutant of *E. coli* (one that is not polar on the downstream minD or minE genes) should generate fewer minicells than a Δ minCDE deletion or a deletion of minD only.

Reply: This is a reasonable hypothesis if one assumes that the Min oscillations constitute the sole mechanism of Z-ring positioning. However, other factors seem to be involved, such as nucleoid occlusion and the Ter linkage. The Δ MinC mutant of *E. coli* appears to have the same phenotype as the Δ MinCDE mutant under a microscope. A more quantitative investigation is needed to assess the occurrence of minicells in these two strains. We hope that our in vitro findings will stimulate future in vivo research to challenge the proposed model.

5) The authors should stress that the Min system is not required for FtsZ ring formation or even midcell FtsZ ring formation, as minicell strains still divide at midcell (as well as at the poles) because of nucleoid occlusion and other unknown factors. Fig. 5 and the accompanying text should be modified appropriately.

Reply: We have already mentioned in the Discussion (lines 360-362) that *“in slow growing E. coli cells, condensation of membrane-bound FtsZ filaments into a ring occurs in the absence of the Min system⁵⁸, indicating that other mechanisms guide the ring assembly and its localization.”*

We added a few more sentences to specifically mention minicells (comment #4 above) and nucleoid occlusion (comment #6 below) in lines 358-360 and 362-365.

6) Along the same lines, it should be noted in the Discussion and in the description of the model in Fig. 5 that other factors are important for centering the FtsZ ring in *E. coli* (Ref. 43), including SlmA-mediated nucleoid occlusion. The recent report of SlmA being able to bind to lipid membranes (PMID 32873767) suggests that SlmA on the DNA and the membrane may directly influence FtsZ binding at the membrane and help to focus the FtsZ ring at midcell.

Reply: We added a sentence about SlmA in the revised manuscript, see point above.

7) In the same vein, the *B. subtilis* nucleoid occlusion factor Noc uses an amphipathic helix to bind to

the membrane and sterically hinders FtsZ from assembling into large structures outside of the Nucleoid-delimited zone around midcell (PMID 33531398). This is somewhat analogous to the authors' model for MinDE-mediated FtsZ assembly patterning.

Reply: We agree with the analogy, but we think that the possibility that nucleoid occlusion factors may sterically hinder FtsZ from assembling into a ring is clearer now we have considered comment #6.

8) Throughout the manuscript, starting on line 71, the authors use the term "FtsA-FtsZ filaments" or "co-filaments (e.g. line 23 in the Abstract, line 169 in text, Fig. S3 legend heading, etc.). However, this term is misleading, as in *E. coli*, there is no evidence that FtsA forms filaments that align with FtsZ filaments, particularly at the initial proto-ring stage, which is implied by the terms "filaments" and "co-filaments". On the contrary, it was reported that purified FtsA forms mini-rings on lipid membranes. These mini-rings act as anchors and guiding tracks to align FtsZ protofilaments prior to condensation of FtsZ protofilaments into more bundled forms (PMID 28695917). FtsA mutants that bypass early steps in division form curved and straight filaments instead of mini-rings (PMID 29995995), but it is not clear if these structures are present during the proto-ring stage or only at later stages. Therefore, to be more accurate, this term should be changed to something like "FtsA-anchored FtsZ filaments" or, as described in the Introduction, as "FtsA-FtsZ cytoskeletal structures".

Reply: We agree that some of these terms can be misleading, and we have tried to consistently use "FtsA-anchored FtsZ filaments" or "FtsA-FtsZ cytoskeletal structures" in the revised manuscript.

9) There are several reports of copolymers formed by MinC and MinD, e.g. PMID 25500731, 29610277, although their existence/relevance in cells is controversial. Were copolymers detected in the SLB system? Would the addition of MinC to the MinDE proteins be expected to assemble into copolymers?

Reply: Thank you for pointing to these reports. We did not observe MinC-MinD copolymers in the present study. The presence of MinE in our assays most likely explains why copolymers were not detected, since MinE was shown to disassemble them (PMID 25500731). Also, in general, these MinC-MinD copolymers have a length in the range of nanometers and are typically observed using electron microscopy. This may also explain why we did not detect them with our optical microscopes.

10) The original report of FtsZ counter-oscillating with Min was PMID 15242613.

Reply: We added this reference.

11) Line 292: It should be recalled that Osawa and Erickson showed that FtsZ tethered to the membrane with a hypermorphic mutant of FtsA (FtsA*) could form rings at the center of liposomes

and constrict them (PMID 23776220), so the idea of using FtsA as a natural membrane anchor for FtsZ protofilaments in synthetic “cells” is not new.

Reply: We cited this paper in the revised manuscript on page 7, when introducing our new experiments with FtsA*.

Minor comments:

1) Lines 100-102: it should be mentioned here that chaperones were also added to the in vitro membrane systems.

Reply: Done.

2) Line 104, 123: “ring” or “ring-like” structures are not obvious at all in Fig. 2B, only some structures that are above the background fluorescence. These structures are much clearer in Fig. 2C, although my copy is probably not at sufficiently high resolution to ascertain that they are rings or even ring-like—perhaps string-like would be more appropriate. “Ring” implies closed structures, which are not obvious in these images.

Reply: Although most FtsA-FtsZ structures appear as curved filaments in some images, ring-like structures can also be seen. Given the large wavelength of the Min waves, we imaged large fields-of-views and haven’t really optimized the acquisition settings for smaller cytoskeletal structures. However, previous fluorescence microscopy data, including from our group (ref. 31, Supplementary Fig. 1, where superresolution microscopy was utilized), convincingly showed the formation of closed structures. Therefore, we believe that using terms, such as “ring-like” and “curved filaments” is correct, even if this might not be obvious from the present images.

3) Page 4: regarding the wavelength measurements: the wavelength and oscillation time probably depends a great deal on the MinE concentration relative to MinD. Was this tested?

Reply: Mass spectrometry data suggest that the MinE/MinD ratio ranges from about 0 to 1.6 during a PURE reaction (Figure 1C). We studied Min dynamics over time in combination with gene expression (Figure S5) and calculated the oscillation properties at different time points for which the MinE/MinD ratio could be estimated. We found that the MinDE wavelength and velocity are not influenced (new Fig. S5 D) and that the displacement of the membrane-bound FtsA-FtsZ complex is not impacted once the MinDE waves have developed (Fig. S5 A-C). To expand the range of MinE/MinD ratios, while increasing the absolute amounts of proteins, we repeated the assay described in Fig. 3 A, this time by

supplementing the reaction solution on the SLB with known quantities of purified MinD or MinE. Changing the MinE/MinD ratio did not influence the wave properties (Fig. S4 D).

The absence of changes in MinDE wave properties and displacement of FtsA-anchored FtsZ filaments is consistent with data reported in ref. 27, where different MinD/MinE (10 to 0.1) ratios were shown to not influence pattern dynamics.

4) Line 126-127: Another explanation for the FtsZ-MinC colocalization is that MinC binds to FtsZ directly in two steps (MinCc-FtsZCterm, then MinCn-FtsZ core domain). This is enough of an interaction to expect colocalization in a purified system. It is also consistent with transient localization of oscillating MinC to constricting FtsZ rings in cells (PMID 10540287).

Reply: We included the suggested reference. However, we don't think this represents 'another explanation', as this is exactly what we tried to say: MinC and FtsZ directly interact. We think the confusion comes from the term 'non-membrane-bound FtsZ'. We simply mean that FtsZ is not recruited to the membrane by ZipA or FtsA. For clarity, we rephrased the sentence as: *"The observed colocalization of the MinC and FtsZ patterns was consistent with previous results that indicated weak transient interaction between MinC and FtsZ^{40,41}."*

5) Fig. 3C and E: it is not clear what the scale bars represent.

Reply: We defined the scale bars in the figure caption.

6) Fig. S2 legend: on line 3, change "left" to "bottom".

Reply: Done.

Reviewers' comments:

Reviewer #1 (Remarks to the Author):

This is a revised manuscript reporting that FtsA-FtsZ counter oscillates with MinDE on model membranes in vitro and that MinC is not necessary for this behavior. This is clearly demonstrated. The work is well done and a rather impressive feat. My only concern is the physiological relevance and perhaps there does not have to be any as this is largely an in vitro reconstitution tour de force.

As far as I am aware the phenotype of MinC mutants is the same as that of a MinCDE deletion. This suggests that any function attributable to MinDE is difficult to observe in vivo or at least has not been observed to date. Although the authors are careful in the results section to stick with the observations, in the discussion they suggest that the results have implications for in vivo. However, MinC is absolutely essential in vivo for spatial regulation of Z rings, without MinC Z rings form at cell poles and at midcell a clear indication of the loss of spatial regulation.

What is clearly demonstrated is the effect is on FtsA and appears to have little dependence on FtsZ since FtsA alone counter oscillates with MinDE (Fig. S4C). Since FtsA is thought to form polymers at the membrane this is quite a feat in itself. This is especially true since the turnover of FtsA polymers has never really been demonstrated. In contrast, FtsZ filaments are known to be very dynamic under all conditions. If the behavior of FtsZ is different that that of FtsA alone then this should be highlighted. Otherwise, it would be of interest to track FtsA in the presence of FtsZ to see if FtsA is affected, or does it behave like FtsA in the absence of FtsZ. The fact that the effect is primarily due to an effect on the membrane anchor rather than FtsZ appears supported by the arguments in the discussion (line 304-312). When ZipA is immobile (ref. 39) MinC is required whereas when it is not (this study), MinC is not required.

It is known that the in vitro system has differences from the in vivo (e.g. longer wavelength). Is it possible that the counter oscillation in vitro is due to differences as well? Perhaps the density of MinD on the membrane in vitro may be higher than in vivo and that what one is observing in vitro is a blanket of MinD pushing FtsA out of the way.

Lines 294-303. The difference between the effects of MinDE on FtsZ when using FtsA versus FtsA*. Could this be due to the difference in oligomerization state of FtsA and be independent of FtsZ?

Lines 301-3. While it is true that FtsA* mutants display resistance to excess MinC, FtsA* cells display normal spatial regulation of Z rings indicating that FtsA*/FtsZ respond normally to MinCDE.

Line 349. It would be note that the calculated ratio of FtsZ to MinC is estimated to be about 8:1 so free MinC would have limited effect on the pool of cytosolic FtsZ.

Minor edits

line 42, 'event' should be plural

line 46, replace 'at' with between

line 47, replace 'encouragin'g with restricting

lines 104 and 105, what is the word 'extra' doing here

line 177, change 'into' to in,

line 237, protofilament should be plural

Reviewer #2 (Remarks to the Author):

This manuscript highlights the *in vitro* ability of a MinDE complex to spatially regulate FtsA-FtsZ complexes using both supported lipid bilayers and water-in-oil droplets. MinDE spatially regulated FtsA-FtsZ complexes, which are the primary components of the early assembled division machinery in *E. coli*, in the absence of MinC, the major regulator of FtsZ spatially *in vivo*, through spatial reorganization on a surface and independent of MinC. Overall, the observations and models described in the manuscript are interesting and may be useful to understanding the *in vitro* parameters. While the collected data and observed results are not directly comparable to a physiological intracellular system, the *in vitro* studies nonetheless provide additional context for understanding the functional interactions on a lipid surface. This manuscript further advances the technical capabilities in the field for investigators attempting to reconstitute the division machinery *in vitro*.

The authors have satisfactorily addressed my previous concerns. The additional experiments and text are nice additions to the manuscript and enhance the findings. The statistical analyses and interpretations are adequate and robust.

Reviewer #3 (Remarks to the Author):

The authors have addressed the reviewers' concerns admirably, including new experiments with FtsA* and ZipA membrane anchors. These new data help to strengthen their model.

There are some minor revisions needed.

L 42: should be "events"

L 247: "that" should be "than"

L 250: "less extend" should be "lesser extent"

L 296: There is genetic evidence that FtsA* associates with the membrane more strongly than FtsA (PMID 18181692). This would seem to be another related explanation for why FtsA* is more resistant than FtsA to MinDE-mediated redistribution.

L 300: add ref. 19 to 47 here, as ref. 19 first demonstrated a higher packing density for FtsA* on membranes.

L 307-309: How does the cell-free system allow ZipA to insert its transmembrane domain into the SLB? Unlike membrane-associated proteins with amphipathic helices such as FtsA, it is not clear how an integral membrane protein like ZipA can be inserted using a cell free system. Please explain.

L 328: rewrite as "...asymmetric division, which results in the production of minicells".

L 354: change "interaction" to "binding"

L 556: Does "sequence-optimized" mean "codon-optimized"? Please briefly describe.

L 749: this reference has no author names.

Rebuttal letter - Second revision for:

Manuscript: Min waves without MinC can pattern FtsA-anchored FtsZ filaments on model membranes

Reference: COMMSBIO-21-3395

We thank the Reviewers for their enthusiastic evaluation and for their additional comments that helped us further improve the manuscript. The Referee reports are in **black text** and our point-by-point replies are in **blue text**. Changes in the manuscript (main text only) are in **red text**.

The newly revised manuscript has been uploaded as a separate document.

Reviewer #1 (Remarks to the Author):

This is a revised manuscript reporting that FtsA-FtsZ counter oscillates with MinDE on model membranes in vitro and that MinC is not necessary for this behavior. This is clearly demonstrated. The work is well done and a rather impressive feat. My only concern is the physiological relevance and perhaps there does not have to be any as this is largely an in vitro reconstitution tour de force.

Reply: Thank you for the positive comments. Quoting Reviewer #2, "*While the collected data and observed results are not directly comparable to a physiological intracellular system, the in vitro studies nonetheless provide additional context for understanding the functional interactions on a lipid surface.*" This is a statement that we generally agree with. The new insights gained from in vitro assays contribute to a better understanding of the molecular systems in general, possibly in the physiological context, while we should be mindful that the two settings are not directly comparable.

As far as I am aware the phenotype of MinC mutants is the same as that of a MinCDE deletion. This suggests that any function attributable to MinDE is difficult to observe in vivo or at least has not been observed to date. Although the authors are careful in the results section to stick with the observations, in the discussion they suggest that the results have implications for in vivo. However, MinC is absolutely essential in vivo for spatial regulation of Z rings, without MinC Z rings form at cell poles and at midcell a clear indication of the loss of spatial regulation.

Reply: In the revised manuscript, we have been careful in our interpretations. Yet, as motivated in the former rebuttal letter, we believe that discussing our results in the broader in vivo context is a common and valid practice. Saying that 'MinC is absolutely essential in vivo' is not fully correct. On lines 357–359, we wrote "*in slow-growing E. coli cells, condensation of membrane-bound FtsZ*

filaments into a ring occurs in the absence of the Min system⁵⁸, indicating that other mechanisms guide the ring assembly and its localization." Moreover, on lines 355-357: "*It should be noted that in E. coli, the lack of Min system does not totally hinder division; Z rings can develop both at midcell and at cell poles, culminating in minicells and cells with regular or altered sizes"*. The fact that the Min system is not required for FtsZ ring formation or positioning at midcell, was also pointed out by Referee #3 in the former rebuttal letter (Comment #5).

In the new manuscript, line 325, we slightly rephrased the following sentence to emphasize the hypothetical nature of the scenario: "*The dispensable function of MinC in regulating FtsZ-FtsA dynamics, as reported here, suggests that MinDE oscillations could play a ~~more-central~~ role in patterning critical division..."*.

What is clearly demonstrated is the effect is on FtsA and appears to have little dependence on FtsZ since FtsA alone counter oscillates with MinDE (Fig. S4C). Since FtsA is thought to form polymers at the membrane this is quite a feat in itself. This is especially true since the turnover of FtsA polymers has never really been demonstrated. In contrast, FtsZ filaments are known to be very dynamic under all conditions. If the behavior of FtsZ is different that that of FtsA alone then this should be highlighted. Otherwise, it would be of interest to track FtsA in the presence of FtsZ to see if FtsA is affected, or does it behave like FtsA in the absence of FtsZ. The fact that the effect is primarily due to an effect on the membrane anchor rather than FtsZ appears supported by the arguments in the discussion (line 304-312). When ZipA is immobile (ref. 39) MinC is required whereas when it is not (this study), MinC is not required.

Reply: In both FtsA-FtsZ and FtsA conditions, the proteins counter oscillate the MinDE system and have comparable dynamic properties. We calculated for FtsA-FtsZ a wavelength (MinD signal) of $76 \pm 30 \mu\text{m}$ and a wave velocity of $0.5 \pm 0.2 \mu\text{m s}^{-1}$, while for FtsA alone, we calculated a wavelength (FtsA signal) of $61 \pm 24 \mu\text{m}$ and a wave velocity of $0.3 \pm 0.2 \mu\text{m s}^{-1}$. Although these calculations show no significant differences, we do not exclude that other parameters might differ between the two conditions.

We certainly agree that the nature of the membrane anchor has an influence. We wrote on lines 265-266: "*the nature of the FtsZ membrane anchor affects the strength of responsiveness to the Min surface waves*". However, FtsZ itself may play a role too, since FtsA, in particular, interacts with short FtsZ filaments in bulk, which promotes membrane binding. Importantly, we have been cautious to discuss only the regulation of FtsA- or ZipA-anchored FtsZ filaments, the main observation being that large protein networks interacting with the membrane can also be displaced by MinDE oscillations. The spatial regulation of membrane-anchored FtsZ structures may be caused by an effect of the Min

system exclusively on the membrane-binding protein, as suggested by the reviewer. Nonetheless, the stability of the network, which is governed by the interactions between the membrane-associated protein and FtsZ, appears to also affect the dynamic self-organization.

It is known that the in vitro system has differences from the in vivo (e.g. longer wavelength). Is it possible that the counter oscillation in vitro is due to differences as well? Perhaps the density of MinD on the membrane in vitro may be higher than in vivo and that what one is observing in vitro is a blanket of MinD pushing FtsA out of the way.

Reply: In the revised manuscript, we stressed the differences between in vitro and in vivo. On line 350, we wrote: *"We acknowledge that many physiological factors might influence the dynamic interplay between the Min and FtsA-FtsZ subsystems, which cannot effectively be reproduced in our assays."*

This is an inherent limitation of all in vitro studies. We agree that the density of membrane-bound MinD might be higher on SLBs than in *E. coli* cells, where many other proteins are associated to the inner membrane. The effects of protein density on SLBs were experimentally addressed in ref. 27. The authors demonstrated that the redistribution of membrane-anchored proteins can robustly be seen for MinD/MinE ratios ranging from 10 to 0.1. Counter oscillations were observed at ratios of membrane proteins to MinDE ranging from 30 to 0.1. Moreover, MinDE was also able to dynamically pattern a highly crowded bilayer containing a lipid-anchored streptavidin at a density of 6600 molecules/ μm^2 . These observations, together with the facts that MinCDE counter oscillating with FtsA-FtsZ is demonstrated in both in vivo and vitro settings, and that MinDE act as a general propagating diffusion barrier (ref 27 and 28), support our hypothesis that a similar phenomenon may occur in vivo.

Lines 294-303. The difference between the effects of MinDE on FtsZ when using FtsA versus FtsA*. Could this be due to the difference in oligomerization state of FtsA and be independent of FtsZ?

Reply: Yes, the difference is certainly due to different oligomerization states of FtsA versus FtsA*. However, the dependency of FtsZ is hard to disentangle. For this reason, we have been cautious to discuss only the regulation of FtsA/FtsA*-anchored FtsZ filaments. FtsA* has indeed a reduced propensity to form oligomers compared to FtsA, which improves the lateral interactions between FtsZ protofilaments. Although we cannot completely rule out that FtsA* differently responds to MinDE waves than FtsA with or without FtsZ, we believe that the difference in FtsZ filaments lateral interactions when complexed to FtsA versus FtsA* is the main determinant of the responsiveness to MinDE.

Lines 301-3. While it is true that FtsA* mutants display resistance to excess MinC, FtsA* cells display normal spatial regulation of Z rings indicating that FtsA*/FtsZ respond normally to MinCDE.

Reply: We agree with the remark. FtsA* mutant is resistant to excess MinC, whereas wild-type FtsA is not. The fact that MinC does not enhance the redistribution of FtsA*-FtsZ in our in vitro assays, but does so when using the wild-type FtsA, implies that FtsA* affects the response to MinC. Possible mechanisms are described from line 293.

Line 349. It would be note that the calculated ratio of FtsZ to MinC is estimated to be about 8:1 so free MinC would have limited effect on the pool of cytosolic FtsZ.

Reply: As FtsZ concentration is constant throughout *Escherichia coli* cell cycle (PMID: 28804128, PMID: 19680248 and PMID: 12754232), it appears that the local concentration of FtsZ is more relevant than the total concentration. The robust assembly of the FtsZ ring at midcell causes an increase in the local concentration of FtsZ, lowering the quantity of FtsZ at the cell poles, which reduces the FtsZ:MinC molecular ratio. Therefore, locally, free MinC may influence the pool of cytosolic FtsZ.

Minor edits:

line 42, 'event' should be plural

Reply: Corrected.

line 46, replace 'at' with between

Reply: The Min proteins are found at the poles. The word “between” can be misleading as it may suggest that the proteins are also found at midcell. Therefore, we prefer to keep the original sentence.

line 47, replace 'encouragin'g with restricting

Reply: Corrected.

lines 104 and 105, what is the word 'extra' doing here

Reply: The term "extra" points to the fact that we added the indicated quantity of GTP and ATP on top of the nucleoside triphosphates already present in the PURE system.

line 177, change 'into' to in,

Reply: Changed.

line 237, protofilament should be plural

Reply: Changed.

Reviewer #2 (Remarks to the Author):

This manuscript highlights the in vitro ability of a MinDE complex to spatially regulate FtsA-FtsZ complexes using both supported lipid bilayers and water-in-oil droplets. MinDE spatially regulated FtsA-FtsZ complexes, which are the primary components of the early assembled division machinery in E. coli, in the absence of MinC, the major regulator of FtsZ spatially in vivo, through spatial reorganization on a surface and independent of MinC. Overall, the observations and models described in the manuscript are interesting and may be useful to understanding the in vitro parameters. While the collected data and observed results are not directly comparable to a physiological intracellular system, the in vitro studies nonetheless provide additional context for understanding the functional interactions on a lipid surface. This manuscript further advances the technical capabilities in the field for investigators attempting to reconstitute the division machinery in vitro.

The authors have satisfactorily addressed my previous concerns. The additional experiments and text are nice additions to the manuscript and enhance the findings. The statistical analyses and interpretations are adequate and robust.

Reply: We thank the Referee for the positive comments.

Reviewer #3 (Remarks to the Author):

The authors have addressed the reviewers' concerns admirably, including new experiments with FtsA* and ZipA membrane anchors. These new data help to strengthen their model.

There are some minor revisions needed.

L 42: should be "events"

Reply: Corrected.

L 247: "that" should be "than"

Reply: Corrected.

L 250: "less extend" should be "lesser extent"

Reply: Corrected.

L 296: There is genetic evidence that FtsA* associates with the membrane more strongly than FtsA (PMID 18181692). This would seem to be another related explanation for why FtsA* is more resistant than FtsA to MinDE-mediated redistribution.

Reply: We believe the Reviewer means PMID: 18186792 (the indicated citation is not relevant). In this study, cell fractionation revealed that FtsA* is quite effective in membrane binding, but there was no indication of better binding compared to FtsA. In PMID: 28695917, the authors show that FtsA* is at least as good as FtsA in binding to lipid membranes. More recently, in ref 27, FtsA and FtsA* displayed similar capacity to interact to a membrane, with FtsA* having somewhat lower affinity than FtsA. As we haven't found in the literature a strong support to the higher membrane affinity of FtsA* versus FtsA, we would prefer to not mention this scenario as a possible explanation for the stronger resistance of FtsA*-FtsZ to MinDE-mediated reorganization.

L 300: add ref. 19 to 47 here, as ref. 19 first demonstrated a higher packing density for FtsA* on membranes.

Reply: Added.

L 307-309: How does the cell-free system allow ZipA to insert its transmembrane domain into the SLB? Unlike membrane-associated proteins with amphipathic helices such as FtsA, it is not clear how an integral membrane protein like ZipA can be inserted using a cell free system. Please explain.

Reply: The spontaneous integration of transmembrane domains into supported bilayers is well documented but we do not believe it generally applies to all proteins. For instance, the presence of large cytoplasmic and extracellular domains, or several membrane-spanning domains might impede functional insertion or hamper protein mobility. In our assay, ZipA must spontaneously integrate into the lipid bilayer by its N-terminus transmembrane helix. In a first step, ZipA may interact with the surface of the bilayer via the N-terminal helix which subsequently flips into the membrane. Another possible scenario consists of the co-translational insertion of ZipA. Although we expect most proteins to be expressed (3 hours, test-tube reactions) before addition to the SLB, it could be that a fraction of ZipA is still being produced during incubation to the membrane. In that case, hydrophobic interactions between the lipid bilayer and the transmembrane region of the nascent protein may promote its spontaneous insertion.

L 328: rewrite as "...asymmetric division, which results in the production of minicells".

Reply: Changed.

L 354: change “interaction” to “binding”

Reply: Changed.

L 556: Does “sequence-optimized” mean “codon-optimized”? Please briefly describe.

Reply: We clarified this point: “(sequence-optimized for codon usage, CG content, and 5’ mRNA secondary structures).”

L 749: this reference has no author names.

Reply: We fixed the reference.